# Interplay of SLC33A1-dependent and -independent Golgi sialic acid *O*-acetylation in CASD1 catalysis

Malena Albers[1,13], Lydia Bosse[1,13], Larissa Schröter[1], Anna-Maria T. Junemann[1], Charlotte Rossdam[1,2], Maike Hartmann[1], Melanie Grove[1], Thomas Litfin [3,10], Anna-Sophia Egger [4,11], Marcel Kwiatkowski [4,12], Kathrin Thedieck[4,5,6,7,8], Georg Zocher[9], Falk F. R. Buettner [1,2], Alpeshkumar K. Malde [3], Mark von Itzstein [3] & Martina Mühlenhoff [1] ✉

Sialic acid *O*-acetylation is implicated in the modulation of sialoglycan recognition and ganglioside biology. The sugar modification is catalyzed by CASD1, a Golgi membrane protein that encompasses a luminal catalytic domain and a multipass transmembrane domain. The mechanism of how acetyl-CoA is provided to the Golgi remains poorly understood. Here, we show that the acetyl-CoA transporter SLC33A1 provides acetyl-CoA to the luminal domain of CASD1 and that patient-derived SLC33A1 variants linked to inherited neurodevelopmental and neurodegenerative disorders impair ganglioside 9-*O*-acetylation. Under conditions that enable the formation of 7,9-di-*O*-acetylated sialoglycans, genetic inactivation of SLC33A1 impaired di-*O*-acetylation, but unexpectedly, still enabled mono-*O*-acetylation. Structure prediction and site-directed mutagenesis revealed a second active site in CASD1 that shares striking similarities with the catalytic acetyl-CoA binding transmembrane tunnel of the lysosomal acetyltransferase HGSNAT. Together, our data provide strong evidence that CASD1 has dual functionalities and catalyzes 7,9-di-*O*-acetylation through SLC33A1-dependent luminal acetylation and SLC33A1-independent transmembrane acetylation.

Sialic acids (Sia), a diverse family of nine-carbon α-keto acid sugars, are versatile glycan building blocks in vertebrates. Members of this sugar class are prominently found at the distal ends of N- and O-glycans of glycoproteins and are an integral component of the glycan moiety of an important class of glycosphingolipids, the gangliosides[1,2]. Sialoglycans are involved in numerous molecular interactions, fulfill critical functions in development and immunity, and are frequent targets for pathogens to promote host cell entry[1,3,4]. The major biosynthetic route

[1]Institute of Clinical Biochemistry, Hannover Medical School, Hannover, Germany. [2]Proteomics, Institute for Theoretical Medicine, University of Augsburg, Augsburg, Germany. [3]Institute for Biomedicine and Glycomics, Griffith University, Gold Coast, QLD, Australia. [4]Institute of Biochemistry and Center for Molecular Biosciences Innsbruck, University of Innsbruck, Innsbruck, Austria. [5]Department Metabolism, Senescence and Autophagy, Research Center One Health Ruhr, University Alliance Ruhr and University Hospital Essen, University Duisburg-Essen, Essen, Germany. [6]German Cancer Consortium (DKTK), partner site Essen/Duesseldorf, a partnership between German Cancer Research Center (DKFZ) Heidelberg and University Hospital Essen, Essen, Germany. [7]Center of Medical Biotechnology, Faculty of Biology, University of Duisburg-Essen, Essen, Germany. [8]Westdeutsches Tumorzentrum (WTZ), Essen, Germany. [9]Inter-faculty Institute of Biochemistry, University of Tuebingen, Tuebingen, Germany. [10]Present address: Structural Biology Facility, Mark Wainwright Analytical Centre, University of New South Wales, Sydney, NSW, Australia. [11]Present address: Precision Proteomics Center, Swiss Institute of Allergy and Asthma Research (SIAF), University of Zurich, Davos, Switzerland. [12]Present address: Department of Biomedicine, University of Bergen, Bergen, Norway. [13]These authors contributed equally: Malena Albers, Lydia Bosse. ✉e-mail: muehlenhoff.martina@mh-hannover.de

of Sia starts from *N*-acetylmannosamine and provides 5-*N*-acetylneuraminic acid (Neu5Ac), the most frequent Sia type. Size, shape, and functionality of Neu5Ac is further modulated by a variety of post-synthetic modifications[5]. In humans, the most prevalent modification is *O*-acetylation at the C7 and/or C9 hydroxyl groups of Sia, which typically yields mono-*O*-acetylated Neu5,9Ac$_2$ (9-*O*-Ac Sia) and di-*O*-acetylated Neu5,7,9Ac$_3$ (7,9-*O*-Ac Sia)[6]. Not only can 9-*O*-acetylation mask recognition sites[7–10], it can also create novel sites, which are exploited by viruses that possess specialized Sia-binding proteins for the specific attachment to 9-*O*-acetylated sialoglycans, such as influenza C virus and the human coronaviruses OC43 and HKU1[11–13]. Furthermore, 9-*O*-acetylation also impacts ganglioside biology. For example, the addition of a single acetyl group to the C9 hydroxyl group of the terminal α2,8-linked Sia residue of the ganglioside GD3 counteracts the pro-apoptotic activity of the parental molecule and promotes the survival of various cancer cells[14–16]. Also, in the nervous system, 9-*O*-acetylated gangliosides are implicated in developmental, regenerative, and neuroprotective processes[17–19].

The key enzyme in the biosynthesis of 9-*O*-acetylated sialosides is the sialate *O*-acetyltransferase (SOAT) CAS1 domain containing 1 (CASD1), a Golgi-resident multipass transmembrane (TM) protein composed of a luminal-oriented catalytic domain (LCD) and a C-terminal transmembrane region (CTMR) of unknown function[20,21]. In vitro studies with the isolated LCD demonstrated transfer of acetyl groups from acetyl-coenzyme A (acetyl-CoA) to cytidine 5′-monophosphate (CMP)-activated Sia (CMP-Sia)[20], the donor substrate of the Golgi-resident sialyltransferases. This indicates that *O*-acetylation occurs prior to sialylation and that the LCD depends on luminal acetyl-CoA. Since acetyl-CoA is membrane impermeable and not known to be produced within the Golgi, a transport system is required that provides acetyl-CoA to the Golgi lumen[22].

In 1997, a putative acetyl-CoA transporter cDNA was identified by its ability to increase ganglioside 9-*O*-acetylation upon transfection-mediated expression[23]. The encoded protein, solute carrier family 33 member A1 (SLC33A1; also referred to as AT-1 or ACATN), shares sequence similarity with transporters of the major facilitator superfamily (MFS)[24]. In its solubilized state, the transporter exists as a dimer and exhibits functional characteristics consistent with an acetyl-CoA/CoA antiporter[25,26]. SLC33A1 was initially found in the ER[23,25] and it remained unclear whether it contributes to Golgi acetylation only upon overexpression. Subsequent studies led to the discovery of ER-based Nε-lysine acetylation, implicated in protein quality control and regulation of autophagy[27–29], and a potential Golgi function of SLC33A1 was not further pursued. Gene silencing indicated that SLC33A1 is important for cell viability[25], which may explain why knockout (KO) cells to dissect SLC33A1-dependent processes are still largely lacking. However, a comprehensive understanding of the SLC33A1 acetylation network is of medical relevance, given that genetic dysfunction of SLC33A1 in humans is associated with neurodevelopmental and neurodegenerative disorders. The most devastating type is the autosomal recessive Huppke-Brendel syndrome (HBS), a childhood-onset disorder characterized by profound psychomotor retardation, congenital cataracts, hearing loss, cerebellar atrophy, hypomyelination, and premature death[30,31]. *SLC33A1* mutations acquired through dominant inheritance have been identified in patients affected by spastic paraplegia type 42 (SPG42)[32] and late-onset cerebellar ataxia (ATX)[33]. The effect of disease-causing SLC33A1 mutations on Sia *O*-acetylation is not known.

To date, the precise mechanism by which acetyl groups are delivered to the Golgi lumen and the specific contribution of SLC33A1 for Golgi Sia *O*-acetylation remain elusive. By generating a panel of KO cells, we show here that SLC33A1 is essential for the formation of 9-*O*-acetylated GD3 (9-*O*-Ac-GD3) and that HBS-, SPG42- and ATX-derived SLC33A1 mutations impair ganglioside 9-*O*-acetylation. Moreover, we demonstrate that the O-glycan-specific sialyltransferase ST8SIA6 drives the formation of 7,9-*O*-acetylated sialoglycans in a process that depends on CASD1 but relies only partially on SLC33A1. By combining structure prediction, site-directed mutagenesis and molecular dynamics simulations, we show that the CTMR of CASD1 functions as a TM SOAT with direct access to cytosolic acetyl-CoA. Taken together, we establish a two-catalytic sites model for di-*O*-acetylation involving SLC33A1-dependent luminal Sia *O*-acetylation by the LCD of CASD1 and SLC33A1-independent TM Sia *O*-acetylation by the CTMR of CASD1.

## Results

### SLC33A1 is required for the formation of 9-*O*-acetylated GD3 in HAP1 cells

To define the role of the acetyl-CoA transporter SLC33A1 in Sia 9-*O*-acetylation, we generated *SLC33A1*-deficient cells (Δ*SLC33A1*) by CRISPR/Cas9. Initially, we targeted *SLC33A1* in the near-haploid human cell line HAP1 by introducing a microdeletion in exon 1, leading to a premature stop codon and a massive truncation of the translation product (Supplementary Fig. 1). As a functional readout for Sia 9-*O*-acetylation, we assessed the formation of 9-*O*-Ac-GD3 by immunofluorescence staining with two antibodies that allow discrimination between the 9-*O*- and the non-*O*-acetylated form of GD3 (Fig. 1a, b). The synthesis of GD3 from the precursor ganglioside GM3 is catalyzed by the sialyltransferase ST8SIA1, which extends the glycan moiety of GM3 by an α2,8-linked Sia residue that can carry an *O*-acetyl group at position C9 (Fig. 1c). When expressed in parental HAP1 cells (HAP1-WT), ST8SIA1 induced the de novo display of both GD3 and 9-*O*-Ac-GD3, as shown previously[20] and in Fig. 1a. Expression of ST8SIA1 in HAP1-Δ*SLC33A1* cells yielded GD3, but not its 9-*O*-acetylated counterpart (Fig. 1b; left panel). The defect was rescued by the expression of human SLC33A1 (Fig. 1b, top right image), providing evidence that SLC33A1 is a crucial component of the Golgi-resident *O*-acetylation machinery. Albeit ST8SIA1 expression in HAP1 cells resulted in clusters of GD3- and 9-*O*-Ac-GD3 positive cells as shown in Fig. 1a, their overall yields were surprisingly low, as highlighted by flow cytometry (Supplementary Fig. 2a). Using glycosphingolipid profiling by multiplexed capillary gel electrophoresis with laser-induced fluorescence detection (xCGE-LIF; see schematic representation in Fig. 1d), we identified globotriaosylceramide (Gb3Cer) as the major glycosphingolipid of HAP1 cells and uncovered that GM3 was present only as a minor component (Fig. 1e). This indicates that HAP1 cells convert lactosylceramide, the shared precursor of GM3 and Gb3Cer, predominantly to Gb3Cer, which leads to the observed shortage in the GD3 precursor GM3. Profiling of Chinese hamster ovary (CHO) cells, another cell type known to produce 9-*O*-Ac-GD3 upon ST8SIA1 expression[21,34], revealed that they convert lactosylceramide exclusively to GM3 (Fig. 1f). Consistent with this, transient expression of ST8SIA1 in CHO cells yielded over 80% GD3-positive and 46% 9-*O*-Ac-GD3-positive cells (Supplementary Fig. 2b), which prompted us to generate *Slc33a1*-deficient CHO cells as an alternative model to study SLC33A1 functions in a cellular context.

### Loss of *Slc33a1* abolishes 9-*O*-acetylation of GD3 and GD2 in CHO cells

Human and hamster SLC33A1 share 95% protein sequence identity (Supplementary Fig. 3) and the hamster *Slc33a1* gene has exon-intron boundaries essentially identical to the human gene (schematically depicted in Fig. 2a). To inactivate *Slc33a1* in CHO cells, we used three independent CRISPR/Cas9 approaches. Two single guide RNA (gRNA) approaches yielded mutant clones that harbor frameshift-inducing microindels in either exon 1 (Ex1 fs) or exon 3 (Ex3 fs) of the *Slc33a1* gene, while co-application of two gRNAs resulted in an 11.8 kb-deletion spanning exon 1 through 5 (Ex1-5 del). Full details on the mutations are provided in Supplementary Fig. 4. Using total ganglioside extracts of CHO-WT and mutant cells, we examined the ST8SIA1-induced

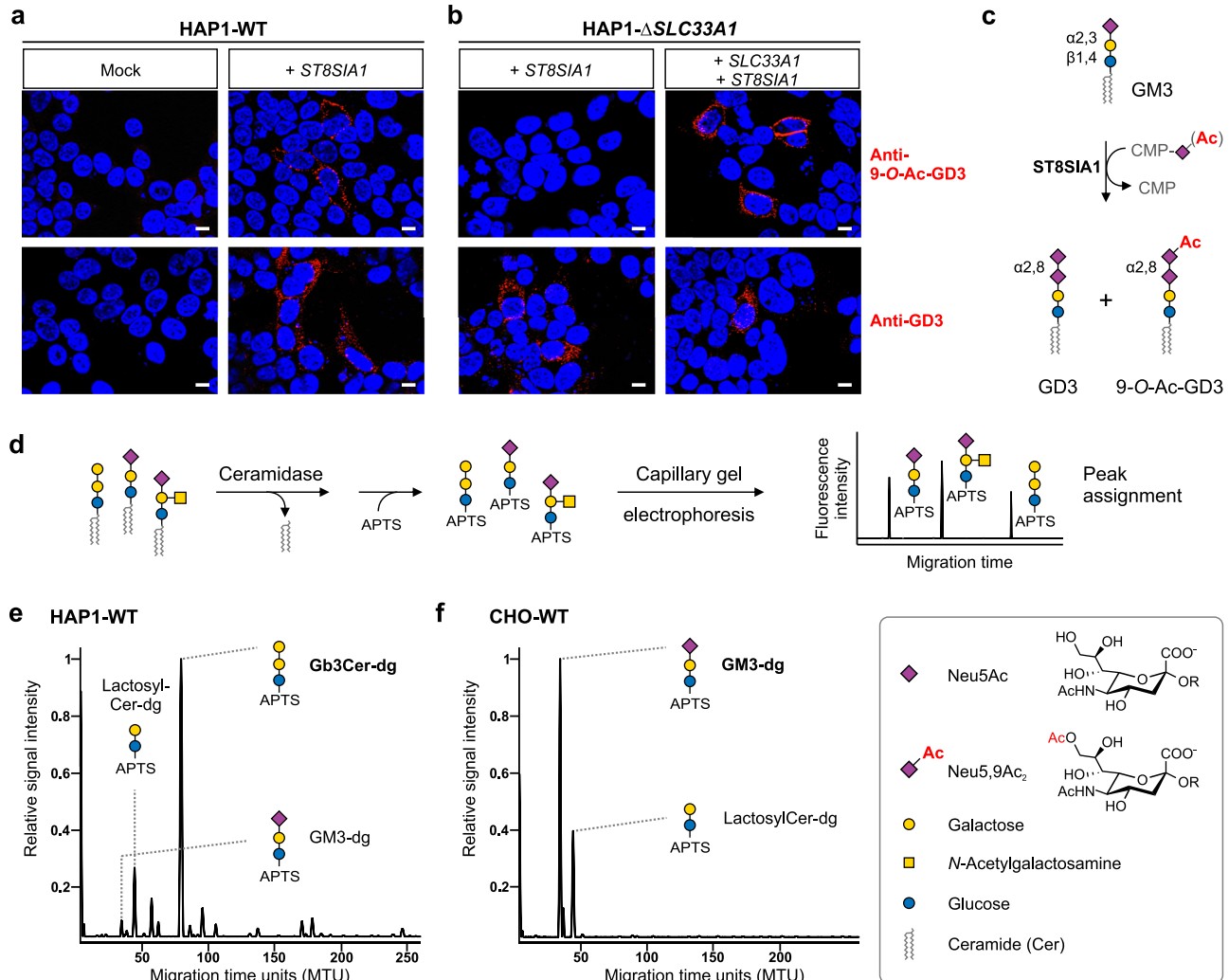

**Fig. 1 | SLC33A1 is required for the 9-O-acetylation of GD3 in HAP1 cells.**
Immunofluorescence microscopy analysis of HAP1-WT (**a**) and HAP1-ΔSLC33A1 cells (**b**) transiently expressing human *SLC33A1* and/or human *ST8SIA1*. Cells were stained with anti-9-O-Ac-GD3 mAb UM4D4 (red, upper panel) or anti-GD3 mAb R24 (red, lower panel). Nuclei were counterstained with DAPI (blue). Cells transfected with an empty vector (Mock) were used as controls. Representative images from one of three experiments are shown. Scale bar, 10 μm. **c** Scheme depicting the ST8SIA1-catalyzed conversion of GM3 to GD3 with concomitant formation of 9-O-acetylated GD3. **d** Schematic work-flow of GSL-derived glycan profiling by xCGE-LIF. Profiles of GSL-derived glycans of HAP1 (**e**) and CHO cells (**f**). Migration times of defined glycan structures were used to assign peaks[65]. APTS 8-aminopyrene-1,3,6-trisulfonic acid trisodium, xCGE-LIF multiplexed capillary gel electrophoresis with laser-induced fluorescence detection, DAPI 4′,6-diamidino-2-phenylindole, dg derived glycan, GSL glycosphingolipid.

formation of GD3 and 9-O-Ac-GD3 by thin-layer chromatography (TLC) followed by antibody overlay (immuno-TLC). An intensely stained GD3 doublet was detected in the extracts of all *ST8SIA1*-transfected cells, irrespective of the underlying genotype (Fig. 2b, lower panel). A 9-O-Ac-GD3 doublet was found in the ganglioside fraction of *ST8SIA1*-expressing CHO-WT cells, but not in the corresponding fraction of the *Slc33a1*-KO cells (Fig. 2b, upper panel). In each of the three CHO-Δ*Slc33a1* clones, the selective loss of 9-O-Ac-GD3 was rescued by transfection-mediated expression of human *SLC33A1* (Fig. 2b, Δ*Slc33a1*, panel '+ SLC33A1'), validating that the acetyl-CoA transporter is an essential component in the synthesis of 9-O-Ac-GD3.

GD3 is a precursor of GD2 and other b-series gangliosides for which 9-O-acetylated forms have been described[35]. As depicted schematically in Fig. 2d, GD2 is synthesized from GM3 by the sequential action of ST8SIA1 and the β4-*N*-acetylgalactosaminyltransferase 1 (B4GALNT1). In CHO cells, co-expression of these two enzymes results in the formation of GD2 and 9-O-Ac-GD2, as shown previously[21] and in Fig. 2c (see WT panel). When we co-expressed ST8SIA1 and B4GALNT1 in *Slc33a1*-KO cells, however, we detected GD2 but not the

corresponding 9-O-acetylated form, as shown exemplarily in Fig. 2c for the CHO-Δ*Slc33a1* clone "Ex1-5 del". The O-acetylation defect was rescued by additional expression of *SLC33A1* (Fig. 2c, far right lane of the top TLC plate). Thus, while genetic inactivation of *Slc33a1* did not affect the synthesis of GD3 and GD2 per se, it fully abolished their modification by 9-O-acetylation.

To rule out indirect effects of *Slc33a1*-deficiency on cell homeostasis, we assessed the metabolic activity of WT and mutant cells in a tetrazolium-reduction assay (Fig. 2e). CHO-Δ*Casd1* cells, which lack Sia 9-O-acetylation but harbor a functional *Slc33a1* gene, were included as an additional control. Over a period of 72 h, we observed no obvious differences between WT and *Slc33a1* KO cells, and after 96 h, only one out of three *Slc33a1*-deficient clones, namely "Ex1 fs", showed a statistically significant reduction in metabolic activity when compared to the WT control. This argues for a clonal difference rather than a specific effect due to *Slc33a1*-deficiency, and we selected clone "Ex1-5 del" for further analyses. Quantification of acetyl-CoA by liquid chromatography–mass spectrometry (LC-MS) revealed elevated levels of this central metabolite in CHO-Δ*Slc33a1* cells compared to control

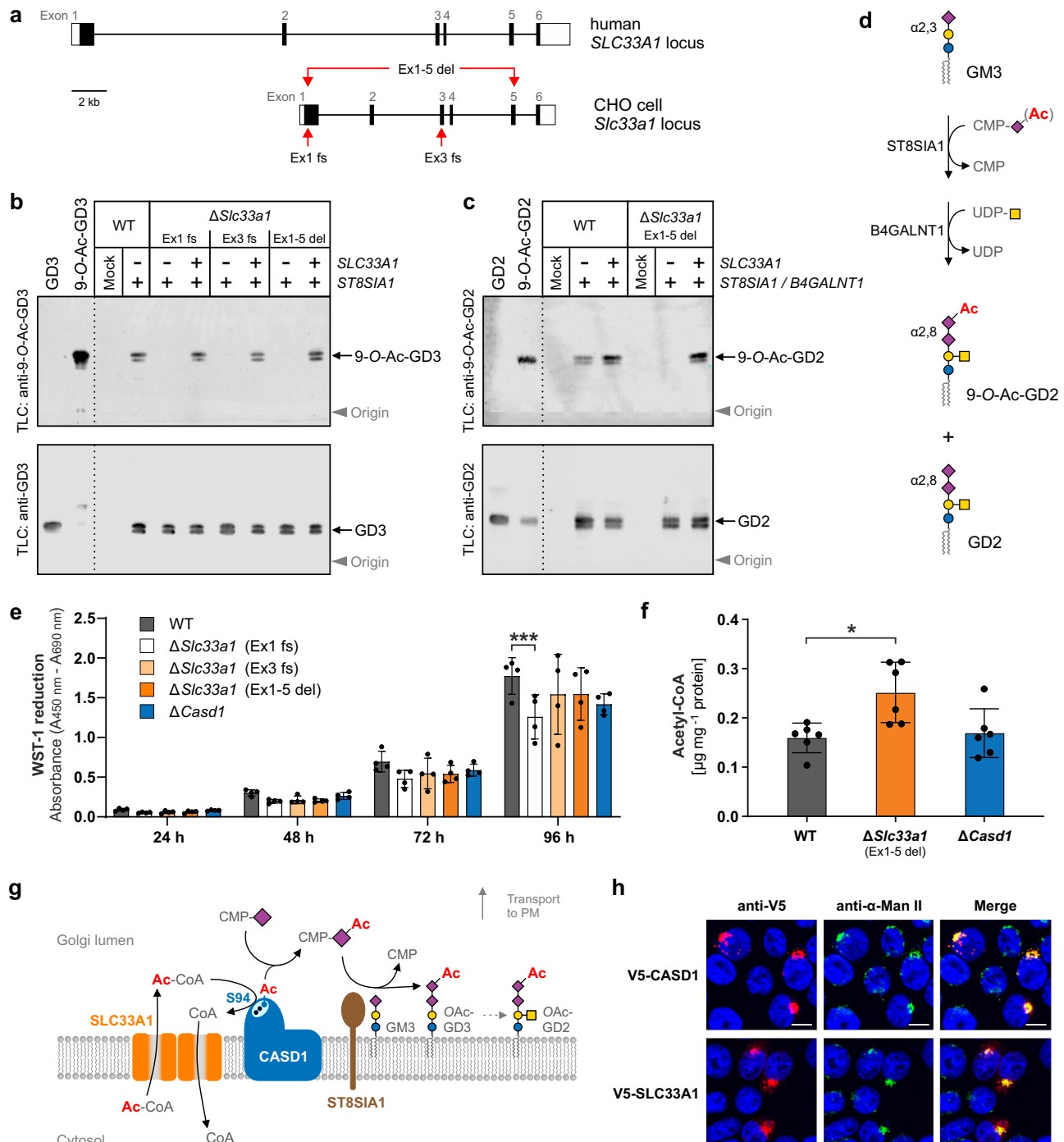

**Fig. 2 | Selective inactivation of *Slc33a1* in CHO cells abolishes the formation of 9-*O*-acetylated GD3 and GD2. a** Scheme of the *SLC33A1* locus in humans and CHO cells, highlighting CRISPR/Cas target sites (red arrows). **b** Immuno-TLC analysis of GD3 and 9-*O*-Ac-GD3 in CHO-WT and CHO-Δ*Slc33a1* cells expressing *ST8SIA1*, with or without exogenous human *SLC33A1*. Total gangliosides were separated by TLC and probed with anti-GD3 (R24) or anti-9-*O*-Ac-GD3 (UM4D4) antibodies. Ganglioside bands appear as doublets due to ceramide structural heterogeneity. Purified GD3 and in vitro 9-*O*-acetyled GD3 were used as standards. Representative plates from one of three independent experiments are shown. **c** Immuno-TLC analysis of GD2 and 9-*O*-Ac-GD2 in CHO-WT and CHO-Δ*Slc33a1* cells expressing *ST8SIA1* and *B4GALNT1*, with and without human *SLC33A1*. Plates were stained with anti-GD2 (ME361) or anti-9-*O*-Ac-GD2 (8B6) antibodies. GD2 and in vitro *O*-acetylated GD2 (containing residual non-*O*-acetylated GD2) were used as standards. Representative plates from one of three independent experiments are shown. **d** Biosynthesis of GD2 from GM3. **e** Cell growth and viability assessed by WST-1 assay. Mean ± s.d., two-way ANOVA with Bonferroni's post-hoc test. *N* = 4 independent experiments,

triplicate measurements. *** $p = 0.00036$. **f** Acetyl-CoA levels measured 24 h post-transfection. Mean ± s.d., one-way ANOVA with Tukey's test. *N* = 6 biological replicates. *$p = 0.0129$. **g** Schematic model of the concerted action of SLC33A1, CASD1 and ST8SIA1 in the formation of 9-*O*-Ac-GD3 and 9-*O*-Ac-GD2. The luminal catalytic site of CASD1 is shown in cyan, with the catalytic triad residues S94 (blue dot), H273, and D270 (black dots). The acetyl-enzyme intermediate, formed via nucleophilic attack by S94[20], is depicted as a blue dot bearing the acetyl group (Ac, red). **h** Golgi localization of CASD1 and SLC33A1. V5-tagged proteins in permeabilized CHO-WT cells were co-stained with anti-V5 (red) and anti-α-mannosidase II (green, Golgi marker). Yellow in merged images indicates colocalization; nuclei stained with DAPI (blue). Representative images from one of three independent experiments are shown. Scale bars: 10 μm. DAPI 4′,6-diamidino-2-phenylindole, Ac-CoA acetyl-coenzyme A, α-ManII α-mannosidase II, PM plasma membrane. Monosaccharide symbols follow the notation in Fig. 1. Source data are provided as a Source data file.

cells (Fig. 2f), which excluded an overall shortage of cellular acetyl-CoA as a possible cause for impaired 9-O-acetylation in Slc33a1-KO cells.

As summarized schematically in Fig. 2g, our data obtained so far suggest that SLC33A1 forms a functional entity with CASD1. To examine whether the two proteins also target to the same compartment, we investigated their subcellular localization by indirect immunofluorescence analysis upon expression of V5-tagged constructs in CHO cells. As shown previously[20], CASD1 colocalizes well with the Golgi marker α-mannosidase II (Fig. 2h, upper panel). Although SLC33A1 displays a more diffuse perinuclear distribution, it nonetheless shows substantial colocalization with the marker protein (Fig. 2h, lower panel), indicating that a significant fraction of the SLC33A1 pool resides in the Golgi apparatus. Our finding contrasts with the strict ER localization observed by subcellular fractionation of CHO cells[25], but is consistent with immunofluorescence microscopy data demonstrating colocalization of SLC33A1 with the Golgi marker proteins giantin and syntaxin 6 in MCF-7 cells[30].

**Patient-derived mutations in *SLC33A1* impact the formation of 9-O-Ac-GD3.** After the successful generation of *Slc33a1*-KO cells, we established a complementation approach to evaluate the functional consequences of patient-derived SLC33A1 variants. As a proof-of-concept, we analyzed p.A110P and p.Y366* underlying childhood onset HBS[30], p.S113R causing SPG42[32], and p.G509S, which has been identified in a patient with late-onset cerebellar ATX[33] (see Supplementary Table 1 for details on all patient-derived SLC33A1 mutations known so far). Each variant was co-expressed with ST8SIA1 in CHO-Δ*Slc33a1* cells, and their capacity to restore GD3 9-O-acetylation was assessed by immuno-TLC. As seen before, expression of SLC33A1-WT readily complemented the Δ*Slc33a1* cells, as indicated by the appearance of a prominent 9-O-Ac-GD3 signal (Fig. 3a, upper panel, +WT/gray box). In contrast, hardly any rescue was achieved with the two HBS variants p.A110P and p.Y366* (blue boxes), indicating that the underlying mutations severely impair SLC33A1 function. The SPG42 variant p.S113R (red box) allowed limited formation of 9-O-Ac-GD3, and the highest complementation capacity was seen for the ATX variant p.G509S (yellow box). For all set-ups, prominent GD3 signals were observed (Fig. 3a, lower panel), validating the functional expression of ST8SIA1 and excluding limitations in the general ganglioside biosynthesis. Detection of 9-O-Ac-GD3 by flow cytometry revealed that each of the analyzed patient-derived variants had a statistically significant lower complementation capacity compared to SLC33A1-WT (Fig. 3b, upper panel; see Supplementary Fig. 5 for gating strategy). Notably, the HBS variants p.A110P and p.Y366* reached only 11.2% and 3.5%, respectively, of the complementation capacity of SLC33A1-WT, whereas a higher restoration capacity of 32% and 57% was seen for p.S113R and p.G509S, respectively. As confirmed by Western blot analysis, all transporter variants were expressed, including the C-terminally truncated nonsense variant p.Y366*, which migrated faster in the SDS-PAGE compared to full-length SLC33A1 (Fig. 3b, lower panel), as shown previously[30]. The anti-Flag antibody used showed two non-specific bands (marked by asterisks in Fig. 3b), with the lower band migrating close to the truncated variant p.Y366*. For clearer visualization, we analyzed additionally V5-tagged constructs, as shown in Supplementary Fig. 6. Together, our data indicate that each of the analyzed mutations significantly impaired SLC33A1 function, with the HBS-derived mutations having the most severe effect.

**Structural analysis of pathogenic SLC33A1 variants**
During the revision of this manuscript, Zhou and co-workers solved a cryo-electron microscopy (cryo-EM) structure of human SLC33A1 in complex with acetyl-CoA[36]. The structure revealed a canonical MFS-fold with 12 transmembrane helices (TMH) arranged as two distinct N- and C-terminal 6-TMH bundles (see Fig. 3c, f for illustration). Among the known patient-derived mutations, Zhou et al. analyzed only the

S113R variant, revealing that residue S113 lies in TMH2 near the substrate cavity and suggesting that substitution of S113 with a bulky arginine residue may interfere with acetyl-CoA binding[36]. In contrast to an earlier report that described the p.S113R variant as inactive[24], Zhou et al. reported substantial residual transport activity[36], a finding that matches well with the partial complementation activity observed for p.S113R in our cellular approach (Fig. 3b).

To obtain structural insights into the other patient-derived mutations analyzed in our study, we mapped the A110P, G509S, and Y366* variants onto the SLC33A1 structure. Residue A110 locates to TMH2 and forms an intrahelical hydrogen bond with Y106 (Fig. 3c, d). Since proline lacks the backbone amide proton required for such bonding, substituting A110 by proline likely induces a helix kink that affects the positioning of Y106. The resolved cryo-EM structure of SLC33A1 adopts an inward-facing conformation in which the extra-cytosolic site is sealed by coordinated packing of TMH1–2 and TMH7–8, reinforced by a lateral gate formed by TMH2 and TMH11[36]. In this configuration, Y106 occludes the luminal face of the substrate cavity (Supplementary Fig. 7) and links the luminal ends of TMH2 and TMH11 through a backbone-side chain hydrogen bond with D472 (Fig. 3d). The A110P substitution likely perturbs the helical geometry of TMH2, displacing Y106 and disrupting extra-cytosolic closure—an effect consistent with the pronounced loss of complementation activity observed for the HBS variant p.A110P (Fig. 3b).

The ATX mutation G509S affects a highly conserved residue (Supplementary Fig. 8) that locates to the luminal end of TMH12, which closely approaches the tip of TMH11 (Fig. 3e). As the smallest amino acid, a glycine permits short interhelical distances and the formation of interhelical backbone-to-backbone Cα-H···O=C hydrogen bonds[37]. Inspection of the cryo-EM structure revealed that the two Cα-H atoms of G509 are in hydrogen bond distance to the backbone carbonyl oxygen atoms of V471 (3.0 Å) and L474 (2.6 Å) of TMH12 (Fig. 3e). Substitution of G509 by a larger serine residue may disrupt this tight arrangement but whether this could affect the architecture of the adjacent luminal loop remains unclear, as this loop is not resolved in the cryo-EM structure[36]. Finally, we located the truncation site of the HBS variant p.Y366* to the C-terminal end of TMH8 (see Fig. 3f for schematic representation). The resulting loss of two-thirds of the C-terminal 6-TMH bundle likely renders this variant non-functional, consistent with its failure to rescue the SLC33A1 knockout (Fig. 3a, b).

**Loss of SLC33A1 impairs 7,9-di-O-acetylation of Golgi-localized sialoglycoproteins.** Previously, we and others found that many mammalian cell lines display O-acetylated sialoglycans in the Golgi even if significant cell surface O-acetylation is missing[20,38]. For CHO cells, we demonstrated that the Golgi-confined O-Ac-Sia residues are predominantly 7,9-di-O-acetylated and attached to N- and O-glycans of so far uncharacterized glycoproteins[20,39]. Having CHO-Δ*Slc33a1* cells in hand, we examined whether SLC33A1-deficiency prevents the O-acetylation of Golgi-confined sialoglycans. As analytical tools, we used virolectins, which are based on the hemagglutinin-esterase (HE) proteins of viruses that use 9-O-acetylated sialoglycans as host cell receptors. Specifically, we used HEs derived from influenza C virus (ICV) and bovine coronavirus (BCoV), which were produced as soluble Fc-chimera of HE variants with a genetically inactivated esterase active site (HE⁰-Fc)[20,39]. The resulting ICV-HE⁰-Fc construct recognizes 9-O-Ac-Sia, whereas the BCoV-derived probe largely prefers 7,9-O-Ac-Sia and shows only limited binding affinity toward 9-O-acetylated ligands[11,39]. In line with our previous data[39], staining of permeabilized CHO-WT cells with BCoV-HE⁰-Fc yielded bright perinuclear signals in virtually all cells, whereas the signals obtained with ICV-HE⁰-Fc were less frequent and mostly of weaker intensity (Fig. 4a, CHO-WT panel). Golgi localization of the intracellular virolectin ligands was confirmed by co-staining with the Golgi marker α-mannosidase II (Supplementary

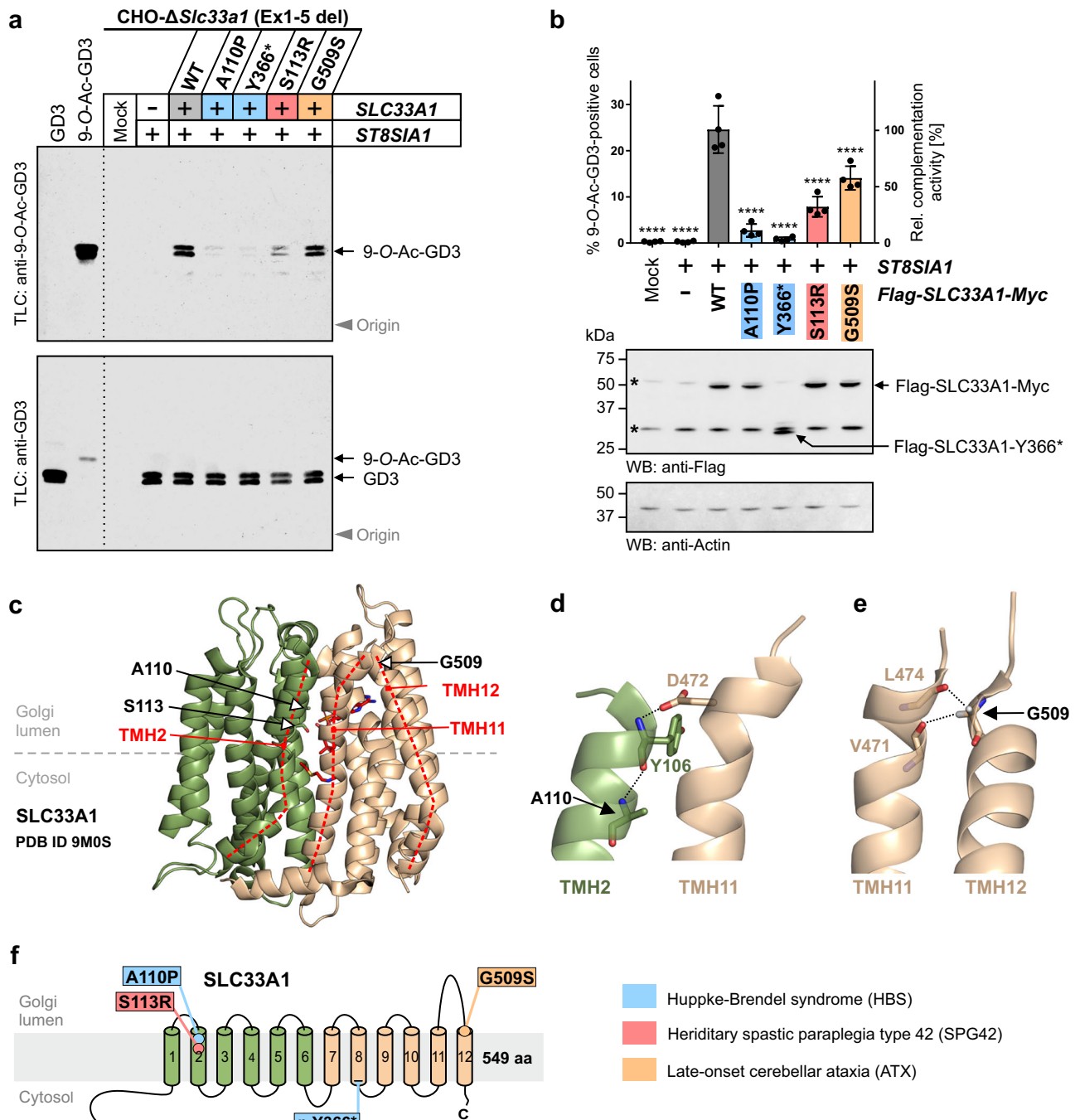

**Fig. 3 | Patient-derived SLC33A1 mutations impair the 9-O-acetylation of GD3.**
**a** Immuno-TLC of extracted gangliosides from CHO-ΔSlc33a1 (Ex1-5 del) cells transfected with a plasmid encoding human V5-tagged ST8SIA1, alone or in combination with a plasmid encoding the indicated N-terminally Flag-tagged SLC33A1 variants. Mock transfected cells were used as negative control. Total gangliosides were extracted and separated by TLC followed by staining with anti-9-O-Ac-GD3 mAb M-T6004 or anti-GD3 mAb R24. Representative plates from one of three independent experiments are shown. **b** Flow cytometric analysis of CHO-ΔSlc33a1 (Ex1-5 del) cells transiently transfected as described in (**a**) and stained with anti-9-O-Ac-GD3 M-T6004. Positive cells were gated as shown in Supplementary Fig. 5. Data are presented as mean ± s.d. of n = 4 independent experiments. One-way ANOVA followed by Bonferroni's post-hoc test. ****p < 0.0001 indicates a significant difference in comparison to ST8SIA1/SLC33A1-WT expressing cells. Western blot analysis was performed to validate the expression of all Flag-tagged SLC33A1 variants. Protein bands representing full-length SLC33A1 and the C-terminally

truncated variant p.Y366* are marked by an arrow. The bands marked by asterisks are already present in the lysate of mock- and ST8SIA1-transfected cells and represent non-specific cross-reactivity of the anti-Flag antibody with endogenous cellular proteins. Actin was used as loading controls. **c** Cartoon representation of the cryo-EM structure of SLC33A1 (PDB ID 9M0S)[36]. The N- and C-terminal six-helix bundles of SLC33A1 are colored in green and beige, respectively. Side chains of the residues in position 110 and 113 are shown as sticks. Dotted red lines visualize the course of the helix axes of TMH2, TMH11, and TMH12. Detail enlargements of the SLC33A1 structure highlighting the positioning of A110 (**d**) and G509 (**e**). Helices are shown as semi-transparent ribbons with selected residues shown in stick representation. Hydrogen bonds are indicated as dotted lines. For clarity, only the main chain atoms of L474 and V471 are depicted. **f** Schematic representation of the membrane topology of SLC33A1. Positions of patient-derived SLC33A1 mutations are indicated. Source data and exact p values are provided as a Source data file.

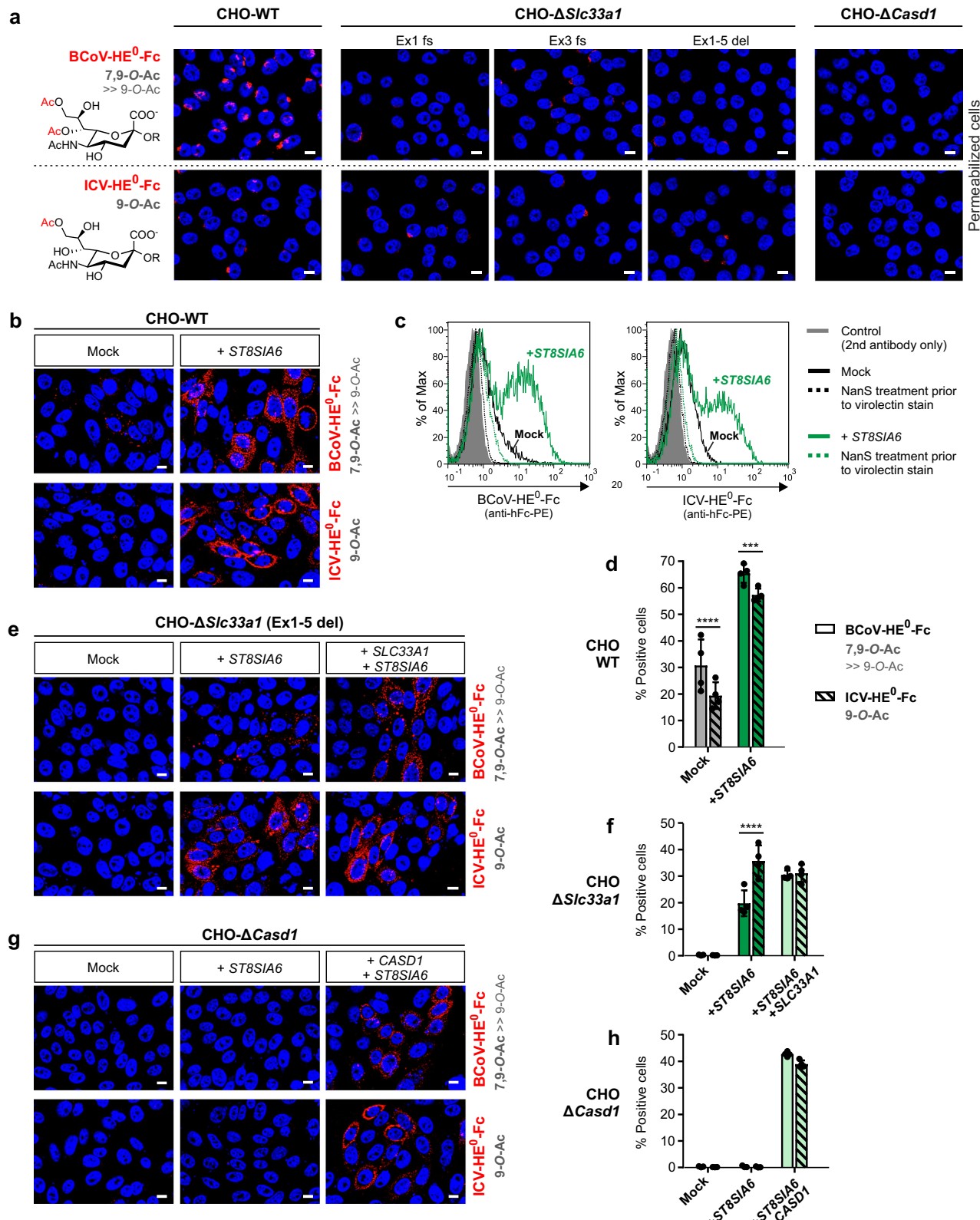

Fig. 9). No virolectin binding was detected in CHO-Δ*Casd1* cells, which were used as a negative control (Fig. 4a, right panel). Intriguingly, all three *Slc33a1*-KO clones showed an intermediate staining pattern. Compared to CHO-WT cells, we observed a severe loss in BCoV-HE[0]-Fc binding, while the respective ICV-HE[0]-Fc binding remained unchanged (Fig. 4a, middle panel). This finding implies that SLC33A1 critically contributes to 7,9-di-*O*-acetylation of Golgi-confined sialoglycans and,

unexpectedly, uncovers that a basal level of mono-*O*-acetylated sialoglycans prevails even in the absence of the acetyl-CoA transporter.

## SLC33A1 is essential but not sufficient for the ST8SIA6-driven formation of 7,9-di-*O*-acetylated sialoglycans

In an attempt to dissect SLC33A1-dependent and -independent forms of Sia *O*-acetylation, we searched for a sialyltransferase that is capable

**Fig. 4 | Impact of SLC33A1 on the formation of 7,9-di-O-acetylated sialosides.**
**a** Fluorescence microscopy images of Triton-permeabilized CHO cells stained with the virolectins BCoV-HE[0]-Fc (7,9-O-Ac-Sia; red) and ICV-HE[0]-Fc (9-O-Ac-Sia; red). Nuclei were counterstained with DAPI (blue). Representative images from one of three independent experiments are shown. Scale bars: 10 μm. Fluorescence microscopy images of non-permeabilized CHO-WT (**b**), CHO-Δ*Slc33a1* (**e**) and CHO-Δ*Casd1* cells (**g**) stained with the indicated virolectins. Prior to staining, cells were transfected with empty vector (Mock) or a plasmid encoding human V5-tagged ST8SIA6, alone or in combination with a plasmid encoding human Flag-tagged SLC33A1 or Myc-tagged CASD1. Nuclei were counterstained with DAPI (blue). Scale bars: 10 μm. The expression of ST8SIA6, SLC33A1 and CASD1 was validated by immunofluorescence microscopy as shown in Supplementary Fig. 12. **c** Virolectin-based flow cytometric analysis of CHO-WT cells before (Mock; black) and after expression of V5-tagged ST8SIA6 (green). Removal of O-acetyl groups by the bacterial sialic acid specific 7,9-O-acetylesterase NanS (dotted lines) was used to validate specific binding of the virolectins BCoV-HE[0]-Fc and ICV-HE[0]-Fc. Mock-transfected cells incubated with secondary antibody only were used as controls (gray fill). Virolectin-based flow cytometric analysis of CHO-WT (**d**), CHO-Δ*Slc33a1* (**f**), and CHO-Δ*Casd1* cells (**h**) transfected with empty vector (Mock) or with a plasmid encoding V5-tagged ST8SIA6 (dark green), alone or in combination with a plasmid encoding SLC33A1 or CASD1 (light green). Results obtained with BCoV-HE[0]-Fc and ICV-HE[0]-Fc are shown as open and striped bars, respectively. Data are presented as mean ± s.d. of n = 4 independent experiments (two-way ANOVA followed by Bonferroni's post-hoc test; ***p < 0.001 and ****p < 0.0001 indicate significant differences between BCoV- and ICV-HE[0]-Fc-stained cells). Dot plots illustrating the gating strategy are shown in Supplementary Fig. 10. See Supplementary Fig. 11 for the western blot analysis confirming that ST8SIA6 expression is not affected by *Slc33a1*-deficiency. BCoV bovine corona virus, Fc constant region of human IgG1, HE[0] hemagglutinin-esterase carrying an alanine exchange of the active site serine of the esterase domain, ICV influenza C virus, PE phycoerythrin. Source data and exact p values are provided as a Source data file.

of producing 7,9-di-O-acetylated sialoglycans destined for cell surface display, as this would facilitate our analysis. Currently, our knowledge on 7,9-di-O-acetylated sialoglycans is sparse, and sialyltransferases responsible for their biosynthesis have not been identified. Based on literature reports, we considered ST8SIA6 as a promising candidate, as it produces an α2,8-linked disialyl motif on O-glycans[40,41], a glycan structure for which both mono- and di-O-acetylated forms have been reported[42,43]. When we expressed ST8SIA6 in CHO-WT cells, we observed a striking increase in cell surface ligands for both ICV- and BCoV-HE[0]-Fc (Fig. 4b, c), indicating that ST8SIA6 induced the de novo display of 9-O- and 7,9-di-O-acetylated sialoglycans on the cell surface. The virolectin binding was sensitive to de-O-acetylation by the sialyl-7,9-O-acetylesterase NanS from *Tannerella forsythia*[39], thereby validating the specificity of the used detection tools (Fig. 4c, solid vs. dashed green tracing).

Mock-transfected CHO-WT cells showed only weak surface binding of the two virolectins. The resulting signals were barely visible by immunofluorescence microscopy (Fig. 4b, left panel), but clearly evident in the flow cytometry histograms (Fig. 4c) by a slight, NanS-sensitive shift (solid vs. dashed black traces) when compared to the secondary antibody control (gray fill). Using a gating strategy that included these weakly stained cells (Supplementary Fig. 10), we observed virolectin binding on up to 30% of the mock transfected CHO-WT cells (Fig. 4d; mock), but on none of the CHO-Δ*Slc33a1* cells (Fig. 4f; mock). When we challenged the *Slc33a1*-deficient cells by expressing ST8SIA6, we detected significant ICV-HE[0]-Fc binding by both fluorescence microscopy (Fig. 4e, lower middle image) and flow cytometry (Fig. 4f, striped dark green column). This indicated that the formation of 9-O-acetylated sialoglycans occurred in an ST8SIA6-dependent but SLC33A1-independent manner. Compared to ST8SIA6-expressing CHO-WT cells, however, we observed a profound loss in BCoV-HE[0]-Fc binding (Fig. 4e, top middle image), indicating diminished 7,9-di-O-acetylation due to *Slc33a1*-deficiency. Exogenous expression of human *SLC33A1* restored BCoV-HE[0]-Fc binding (Fig. 4e; top right image). The complementation led to a ratio of BCoV- to ICV-HE[0]-Fc-positive cells of 1 to 1 (Fig. 4f; solid vs. striped light green columns), whereas without complementation, the ratio was only 0.55 to 1 (solid vs. striped dark green columns). In the latter setting, the actual underlying ratio of 7,9-di-O- to 9-O-acetylated Sia ligands was probably even lower, considering that BCoV-HE[0]-Fc is not fully selective for 7,9-di-O-acetylated Sia and can bind, albeit with low affinity, to 9-O-acetylated ligands[11,39].

In *Casd1*-KO cells, the expression of ST8SIA6 alone was not sufficient to yield O-acetylated sialoglycans (Fig. 4g,h). Co-expression of ST8SIA6 with human CASD1, however, restored Sia O-acetylation (Fig. 4g, right panel) and yielded an almost equal number of BCoV- and ICV-HE[0]-Fc positive cells (Fig. 4h; solid vs. striped light green columns). Western Blot analysis confirmed the expression of ST8SIA6 in WT and

mutant cells (Supplementary Fig. 11), ruling out the possibility that *Slc33a1*- or *Casd1*-deficiency impaired ST8SIA6 expression. The expression and subcellular localization of all transfected constructs were verified by immunofluorescence microscopy (Supplementary Fig. 12). Virolectin stains of permeabilized mock- and ST8SIA6-transfected cells are provided in Supplementary Fig. 13. Collectively, our data showed that the ST8SIA6-driven formation of O-acetylated sialoglycans requires CASD1 but relies only partially on SLC33A1.

## Selective inactivation of the luminal domain of CASD1 recapitulates the phenotype of Δ*Slc33a1* cells

As shown schematically in Fig. 5a, our previous in vitro analysis of the isolated LCD of CASD1 revealed that the luminal domain transfers acetyl groups from acetyl-CoA to CMP-Neu5Ac via an acetyl-enzyme intermediate that involves the hydroxyl group of the catalytic triad nucleophile S94[20]. Introduction of an S94A mutation prevented the formation of the acetyl-enzyme intermediate and thus abolished the SOAT activity of the LCD[20]. The function of the CTMR of CASD1, however, has remained largely unexplored. Based on the AlphaFold2 (AF2) model of CASD1, the CTMR encompasses 14 TMHs that are arranged around a central pore (Fig. 5b). This raises the possibility that the CTMR functions as a membrane SOAT with direct access to cytosolic acetyl groups. If CASD1 comprises two SOAT domains, with only the LCD being dependent on SLC33A1-mediated acetyl-CoA influx, we can expect that inactivation of the LCD and SLC33A1-deficiency will lead to similar outcomes with respect to Sia O-acetylation. To study this experimentally, we complemented CHO-Δ*Casd1* cells with either CASD1-WT or CASD1-S94A. As already shown in Fig. 4f, complementation of CHO-Δ*Casd1* cells with CASD1-WT enables, in the presence of ST8SIA6, the formation of 7,9-di-O- and 9-O-acetylated sialosides (Fig. 5c, left panel). When we repeated this experiment with CASD1-S94A instead of CASD1-WT, we detected a substantial amount of 9-O-acetylated sialosides (Fig. 5c, middle right image), but hardly any 7,9-di-O-acetylated sialosides (Fig. 5c, top right image). This outcome closely resembles the phenotype of ST8SIA6-expressing CHO-Δ*Slc33a1* cells (Fig. 4e, middle panel). Furthermore, complementation of CHO-Δ*Casd1* cells with CASD1-WT, but not CASD1-S94A, enabled the formation of 9-O-Ac-GD3 in the presence of the GD3 synthase ST8SIA1 (Fig. 5d and Supplementary Fig. 14). Thus, selective inactivation of the LCD by introducing an S94A mutation into full-length CASD1 recapitulated both the "impaired 7,9-di-O-acetylation" phenotype (Fig. 4e, middle panel) and the "loss of GD3 9-O-acetylation" phenotype (Fig. 2b) of CHO-Δ*Slc33a1* cells that became apparent in the presence of ST8SIA6 and ST8SIA1, respectively.

## The CTMR of CASD1 functions as transmembrane SOAT

In the Pfam data base[44], the CTMR of CASD1 has been assigned to the AT3 clan (CL0316) since an iterative searching strategy indicated

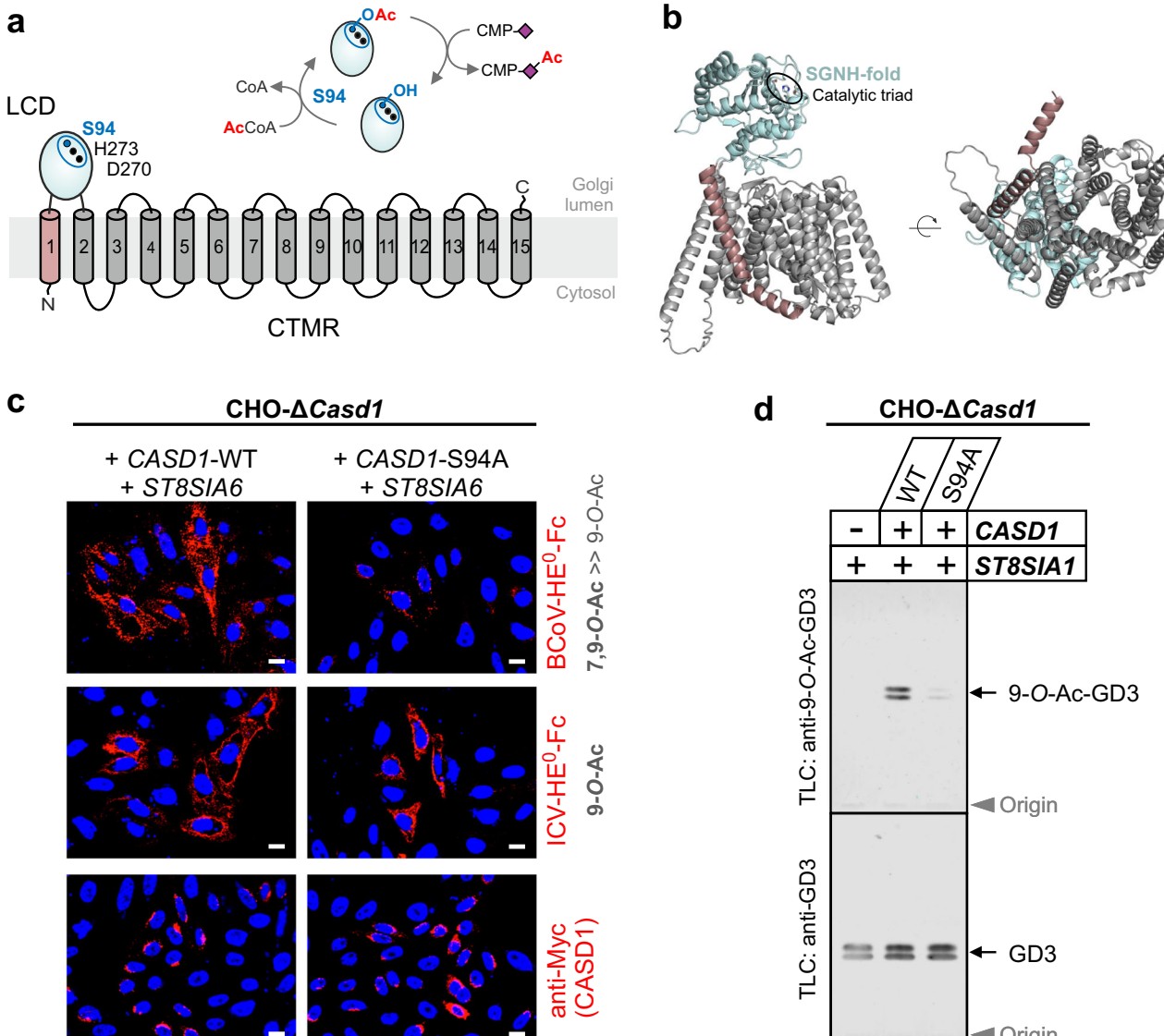

**Fig. 5 | Inactivation of the luminal catalytic domain of CASD1 phenocopies SLC33A1-deficiency. a** Scheme depicting the membrane topology of CASD1 as deduced from experimental data[20] and by inspection of the AF2 model. TMH1, LCD, and CTMR are shown in red, blue and gray, respectively. Residues of the catalytic triad are depicted as dots with the triad nucleophile S94 shown in blue. Inset: Scheme showing the reaction catalyzed by the isolated LCD, i.e., transfer of the acetyl-group from acetyl-CoA to CMP-Neu5Ac via a covalent acetyl-enzyme inter-mediate that involves the hydroxyl group of S94[20]. **b** Cartoon representation of the AF2 structural model of CASD1, displayed as side (left) and bottom view (right). **c** Immunofluorescence microscopy images of CHO-ΔCasd1 cells co-expressing ST8SIA6 with Myc-tagged variants of CASD1-WT (left) or CASD1-S94A (right). Images of cells stained with BCoV- and ICV-HE⁰-Fc are shown in the top and middle panel, respectively. The bottom panel shows images of cells from the same trans-fections that were permeabilized and stained with anti-Myc mAb 9E10 to validate the expression of each of the two CASD1 variants. Representative images from one of three independent experiments are shown. Scale bars: 10 μm. **d** Immuno-TLC of total gangliosides extracted from CHO-ΔCasd1 cells complemented with CASD1-WT or CASD1-S94A. CHO-ΔCasd1 cells were transiently transfected with a plasmid encoding ST8SIA1 alone or in combination with a plasmid encoding either CASD1-WT or CASD1-S94A. Extracted gangliosides were separated by TLC and plates were stained with anti-9-O-Ac-GD3 mAb UM4D4 or with anti-GD3 mAb R24. Repre-sentative plates from one of three independent experiments are shown. CTMR C-terminal transmembrane domain, LCD luminal catalytic domain. Source data are provided as a Source data file.

distant sequence similarity to bacterial sugar acyltransferases of the acyltransferase-3 (acyl_transf_3; AT3) family[45]. As shown in Supple-mentary Table 2, the AT3 clan consists of ten protein families, including the HGSNAT_cat family that comprises the catalytic multi-TM domain of the heparan-α-glucosamide-N-acetyltransferase (HGSNAT). HGSNAT is part of the lysosomal heparan sulfate (HS) degradation machinery and transfers acetyl groups from cytosolic acetyl-CoA to terminal glucosamine residues of luminal HS to enable their subsequent hydrolysis[46]. The recently solved structures of apo- and ligand-bound states of HGSNAT provided detailed insight into the

catalyzed TM acetylation reaction[47–49]. A unique 9-TMH core archi-tecture has been identified that comprises a "scaffold core" and a "catalytic core" (see Fig. 6a, b for schematic representation)[47]. The latter binds cytosolic acetyl-CoA in an extended conformation with the acetyl group pointing towards the luminal-oriented residue H297, which catalyzes the transfer reaction via a ternary complex mechanism[49]. An overlay of the HGSNAT structure with the AF2 model of CASD1 uncovered that TMH5-13 of CASD1 share the 9-TMH core architecture of HGSNAT (Fig. 6b, upper panel), whereas the remaining protein parts lack structural homology (see Supplementary Fig. 15 for

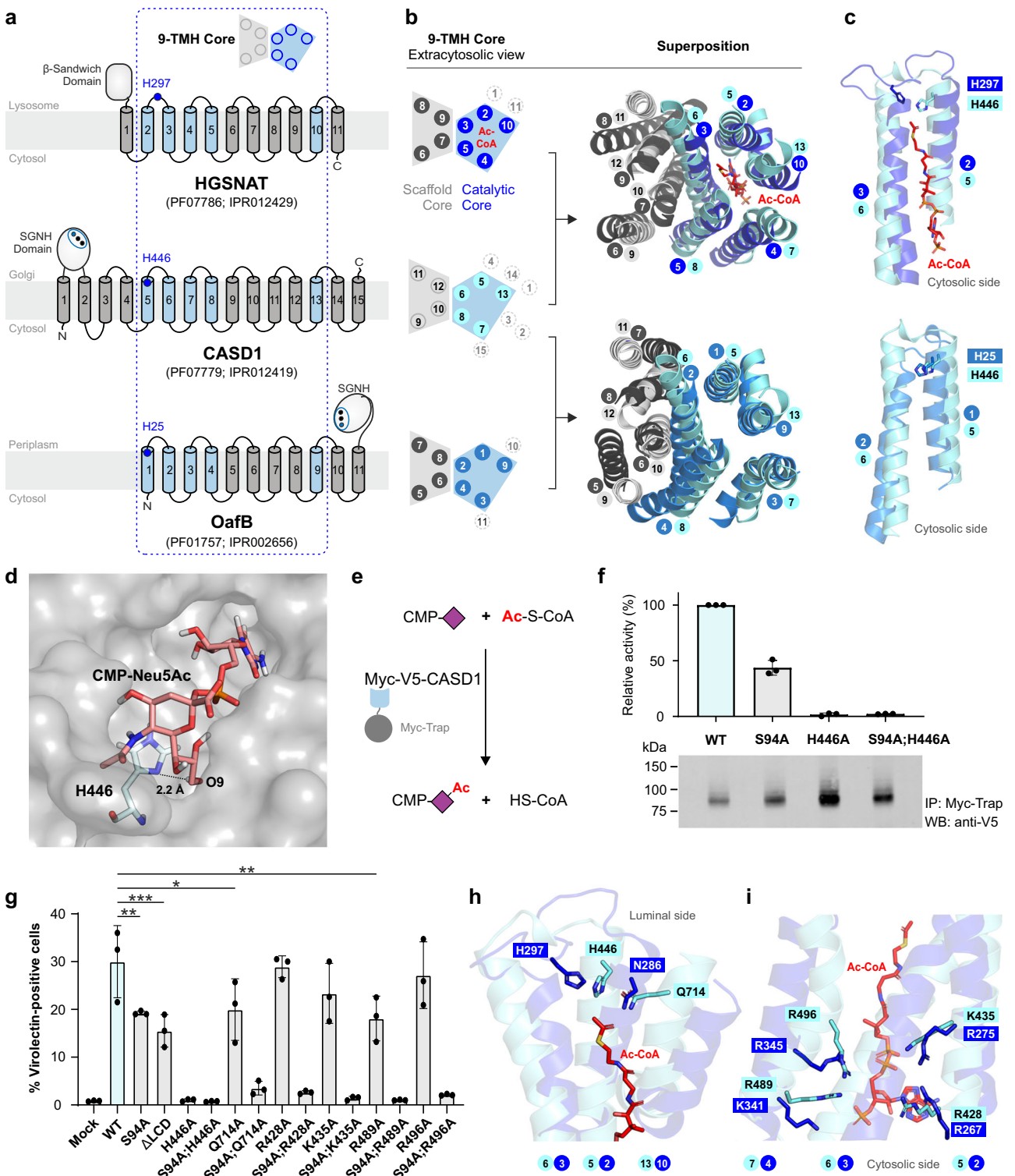

superposition of the full-length proteins). In HGSNAT, the catalytic residue H297 is positioned in a luminal loop (connecting TMH2 and TMH3)[47–49], whereas the equivalent loop in CASD1 (connecting TMH5 and TMH6) lacks a histidine. However, we identified CASD1-H446, located at the luminal end of TMH5, as potential functional equivalent (Fig. 6c, upper panel). This was further substantiated by comparing CASD1 with an AT3 domain-containing *O*-acetyltransferase. To date, there are no experimental structures of AT3 domains, but computational studies on the *Salmonella* O-antigen modifying acetyltransferase OafB suggested that its AT3 domain adopts a novel fold as framework

for TM acyl-group transfer[50]. Our inspection of the AF2 model of OafB revealed a TMH arrangement that is very similar to the 9-TMH core of HGSNAT (Fig. 6b; bottom panel). This includes the splitting of the last "catalytic core"-TMH into two kinked half-helices, a characteristic feature of HGSNAT[47–49] that appears to be conserved in both CASD1 and OafB (Supplementary Fig. 16). By superimposing the AF2 models of CASD1 and OafB, we identified OafB-H25 and CASD1-H446 as positional equivalents (Fig. 6c, bottom panel). Notably, H25 is essential for OafB activity and strictly conserved across all bacterial AT3 domain-containing proteins analyzed so far[51]. Likewise, the residue H446 is

**Fig. 6 | Identification of a second catalytic center in CASD1. a** Membrane topology of HGSNAT, CASD1 and OafB. The conserved 9-TMH core is boxed, with catalytic core TMHs in blue. Pfam/InterPro IDs are provided for the multi-TM domains. **b** Schematic and cartoon representation of the 9-TMH cores with catalytic core TMHs in blue (HGSNAT), cyan (CASD1), and aquamarine (OafB). Scaffold core TMHs are colored in light gray (CASD1) or dark gray (HGSNAT, OafB). Additional TMHs are depicted as dotted circles. Top: AF2 model of CASD1 superimposed on cryo-EM structure of acetyl-CoA-bound HGSNAT (PDB 8TU9), showing acetyl-CoA (red) in stick representation. Bottom: superposition of AF2 models of CASD1 and OafB. Top view from extracytosolic side. **c** Enlarged details (side view), showing the side chains of the catalytic histidine of HGSNAT (H297, which is equivalent to H269 in the numbering system that sets the second methionine as amino acid 1) and of CASD1-H446 and OafB-H25 in stick representation. **d** Snapshot of a 2000 ns MD simulation of CMP-Neu5Ac bound to the C-terminal transmembrane region (CTMR) of CASD1. Protein is shown in semi-transparent surface representation (gray). H446 and CMP-Neu5Ac are shown in stick representation with carbon atoms colored cyan and salmon, respectively. **e** Scheme of on-bead O-acetyltransferase assay. **f** Upper: bar graph of in vitro activity of CASD1-WT (cyan) and CASD1 variants (gray). Mean ± s.d., $n = 3$ independent experiments. Lower: Western blot of affinity-captured CASD1 from one of three experiments. **g** Virolectin-based flow cytometry of CHO-ΔCasd1 cells co-transfected with a plasmid encoding V5-ST8SIA6 along with either empty vector (Mock) or a plasmid encoding one of the indicated Myc-V5-CASD1 variants. Cells stained with ICV-HE⁰-Fc; data represent mean ± s.d. (CASD1-WT, cyan; variants, gray). $N = 3$ independent experiments. One-way ANOVA with Dunnett's test: *$p < 0.05$, **$p < 0.01$, ***$p < 0.001$. **h, i** Details of the superposition of the AF2 model of CASD1 (cyan) and the cryo-EM of HGSNAT (blue) in the acetyl-CoA-bound state (PDB 8TU9), showing acetyl-CoA (red) and selected residues in stick representation. Ac-CoA acetyl-coenzyme A. Source data and exact $p$ values are provided as a Source data file.

conserved not only across CASD1 orthologues of the sialylation competent vertebrate lineage, but also in Cas1p, a Golgi O-acetyltransferase that modifies the capsular polysaccharide glucuronoxylomannan of the pathogenic fungus *Cryptococcus neoformans*[52], as well as in the reduced wall acetylation (RWA) proteins, four Golgi-localized CTMR domain-only proteins that play vital roles in cell wall O-acetylation in Arabidopsis[53] (Supplementary Fig. 17).

Closer examination of the CASD1 model revealed that H446 is situated at the bottom of a luminal pocket that could serve as acceptor binding site. A structure-guided approach was then used to derive an initial CTMR–CMP-Neu5Ac complex, in which we sought to test the stability and orientation preference of the acceptor substrate, using molecular dynamics (MD) simulations. The initial complex, embedded in a 1,2-dipalmitoyl-sn-*glycero*-3-phosphocholine (DPPC) lipid bilayer model, was used to perform three independent MD simulations that commenced from a different configuration and different initial velocity distributions (see Supplementary Fig. 18 for details). The converged structure snapshot after 2000 ns MD simulation is shown in Fig. 6d. From this simulation, we observed that CMP-Neu5Ac occupies the luminal binding pocket in an orientation that positions the 9′-hydroxyl group of CMP-Neu5Ac within hydrogen bonding distance to H446 (see further detail in Supplementary Fig. 18d). This simple MD model supports the notion that the CTMR of CASD1 can accommodate CMP-Neu5Ac in an orientation that allows O-acetylation via H446, possibly by activating the 9′-hydroxyl group for a nucleophilic attack on the carbonyl group of acetyl-CoA. Collectively, these data indicate that the CTMR of CASD1 functions as an SLC33A1-independent TM SOAT with direct access to cytosolic acetyl-CoA.

To directly demonstrate that the CTMR of CASD1 functions as an O-acetyltransferase, we performed an on-bead enzyme assay using detergent-solubilized, Myc-V5-tagged CASD1 immobilized on Myc nanobody-coated beads as enzyme source (Fig. 6e). After incubation with CMP-Neu5Ac and acetyl-CoA, we quantified the acetyltransferase reaction product HS-CoA using a thiol-sensitive dye, as described recently for HGSNAT[48]. Under the used conditions, CASD1-S94A retained over 40% of the WT activity (Fig. 6f). Since this mutant lacks a functional LCD, the remaining SOAT activity can be attributed to the CTMR. Although only the double mutant S94A;H446A—harboring defects in both LCD and CTMR—was expected to be inactive, the single mutant CASD1-H446A likewise showed no detectable activity. Effective enzyme loading was verified by Western blot analysis, as shown exemplarily for one experiment in Fig. 6f and for all biological replicates in Supplementary Fig. 19.

To circumvent potential destabilizing effects of detergent solubilization, we employed cellular complementation using CHO-ΔCasd1 cells to further assess the functional impact of CASD1 mutations. The ability of CASD1 variants to restore 9-O-acetylated sialosides upon co-expression with ST8SIA6 was analyzed by flow cytometry using the ICV-derived virolectin (Fig. 6g; with dot plots and expression controls

shown in Supplementary Figs. 20 and 21, respectively). In the cellular setting, CASD1-S94A retained >60% of the WT complementation activity. To definitively exclude any contribution from the mutated LCD, we replaced the entire domain with a FLAG epitope. The resulting deletion mutant, CASD1-ΔLCD, exhibited complementation activity comparable to CASD1-S94A, indicating that the CTMR alone accounts for the residual activity. CASD1-H446A, like the double mutant CASD1-S94A;H446A, reached no significant activity over the mock control. Likewise, substitution of H446 with alternative residues (H446Y/F/Q/S) invariably abolished complementation activity (Supplementary Fig. 22). In HGSNAT, alanine replacement of N286, positionally equivalent to CASD1-H446 by structural superposition (see Fig. 6h), abolished enzymatic activity and caused abnormal oligomerization[48], highlighting a critical role of this position in protein folding and stability. Moreover, HGSNAT-N286 plays a pivotal functional role by stabilizing the acetyl group of acetyl-CoA within the active site[47–49]. Based on the structural overlay, we identified CASD1-Q714 as putative functional equivalent of HGSNAT-N286, even though the two residues lie in distinct yet adjacent TMHs of the 9-TMH core (Fig. 6h). Q714 is strictly conserved across phylogenetic distant CASD1 orthologues (Supplementary Fig. 17), and its replacement by alanine reduced complementation activity by ~35% (Fig. 6g). This loss increased to >90% when Q714A was combined with the LCD-inactivating S94A mutation (Fig. 6g; CASD1-S94A;Q714A), indicating a functional role of Q714 for CTMR-mediated O-acetylation.

Furthermore, as shown in Fig. 6i, we found that the four CASD1 residues K435, R428, R489 and R496 align structurally with the four positively charged HGSNAT residues that bind the adenosine diphosphate moiety of acetyl-CoA[47–49]. In HGSNAT, these sites are essential (R275 and R267) or contribute significantly (R345 and K341) to enzymatic activity[48]. Consistent with this, alanine substitution of the corresponding CASD1 residues severely impaired complementation activity when combined with the LCD-inactivating S94A mutation (Fig. 6g). In the presence of a functional LCD, however, the loss-of-function effect of the K435A, R428A, R489A and R496A mutations was largely masked by the activity of the LCD. Notably, the four positively charged residues identified as critical for CASD1's CTMR function are not only structurally congruent with the basic acetyl-CoA-binding residues of HGSNAT (Fig. 6i), but also with a quartet of basic residues in the OafB model (Supplementary Fig. 23), further reinforcing the structural homology of the 9-TMH cores across HGSNAT, CASD1 and OafB.

## Discussion

In this study, we provide evidence for the existence of two distinct routes for the translocation of acetyl units across the Golgi membrane, both of which merge in CASD1 catalysis. One route uses the transporter SLC33A1 and provides acetyl-CoA for the LCD of CASD1, whereas the second route follows a TM acetylation mechanism catalyzed by the CTMR of CASD1.

In contrast to the previous assumption that acetyl-CoA reaches the Golgi via vesicular transport from the ER[54], the Golgi localization of SLC33A1 observed in the present study, consistent with findings by Huppke et al.[30], suggests that acetyl-CoA is translocated directly from the cytosol to the Golgi lumen. The positioning of both the acetyl-CoA transporter and the *O*-acetyltransferase within the same compartment ensures the effective utilization of imported acetyl-CoA by the LCD of CASD1. In line with this, pioneering studies on isolated Golgi vesicles, incubated with radioactively labeled acetyl-CoA, provided evidence for the import of the entire acetyl-CoA molecule and found up to 85% of the incorporated radioactivity in *O*-acetylated Sia released from glycans[55].

An important finding of our study was the intriguing structural relationship between CASD1, HGSNAT, and OafB, which led to the identification of a second catalytic site in CASD1 and an SLC33A1-independent route for the translocation of acetyl groups across the Golgi membrane. Our data highlight that both CASD1 and OafB share the 9-TMH core architecture that has been recently and elegantly identified in HGSNAT as the structural basis for acetyl group transfer from cytosolic acetyl-CoA to an extracytosolic sugar moiety[47–49]. The multi-TM domains of CASD1, HGSNAT, and OafB all belong to distinct, but evolutionary related protein families (Supplementary Table 2), indicating that the 9-TMH core became a versatile structural element for several if not for all the sugar acetyltransferases within the AT3 clan. Variations in the interhelical loops, the precise positioning of the catalytic histidine, and the number and position of additional helices may reflect evolutionary adaptations to accommodate different acceptor substrates and to allow TM acetylation in different biological contexts and organisms.

The finding that the CTMR of CASD1 functions as a second catalytic domain clearly advances our understanding of Sia *O*-acetylation. The data presented in this study provide evidence for a mechanistic model (Fig. 7), in which 7,9-di-*O*-acetylation is catalyzed by the concerted action of the LCD and the CTMR of CASD1. Our proposed two-catalytic site model makes it tempting to suggest that the LCD and the CTMR differ in their regioselectivity with respect to the hydroxyl groups in positions C7 and C9 of Sia. Our MD simulations on the CTMR model show binding of CMP-Neu5Ac in an orientation that would allow 9-*O*-, but not 7-*O*-acetylation, suggesting that the CTMR is selective for the C9 hydroxyl group, while the LCD mediates 7-*O*-acetylation. At first glance, this appears puzzling, since our data also demonstrated that the LCD, together with SLC33A1 and ST8SIA1, mediates the formation

of 9-*O*-acetylated gangliosides. However, in light of prior evidence for the cellular occurrence of 7-*O*-acetylated GD3 (7-*O*-Ac-GD3)[56,57], it is plausible that ST8SIA1 uses 7-*O*-acetylated CMP-Sia—produced by the LCD of CASD1—as a donor substrate, thereby initially forming 7-*O*-Ac-GD3. This product can subsequently undergo non-enzymatic conversion to 9-*O*-Ac-GD3, as the 7-*O*-acetyl group is prone to migrate to C9, yielding a thermodynamically more stable primary ester bond[58,59] (see Supplementary Fig. 24 for a schematic of the proposed pathway). Further studies are needed to directly demonstrate the in vitro formation of *O*-acetylated CMP-Sia by the CTMR of CASD1 and to unambiguously define the regioselectivity of the two catalytic domains of CASD1. Because acetyl group migration can mask the initial acetylation site, real-time methods such as nuclear magnetic resonance spectroscopy may be required.

Our data show that SLC33A1 is dispensable for the CTMR-catalyzed Sia *O*-acetylation step, but that it remains a critical component of the Sia *O*-acetylation machinery by providing the donor substrate for the LCD of CASD1. As a gatekeeper for acetyl-CoA imported into the Golgi lumen, SLC33A1 controls LCD catalyzed mono-*O*-acetylation as well as LCD-CTMR co-catalyzed di-*O*-acetylation. Accordingly, genetic inactivation of SLC33A1 abolished the formation of mono-*O*-acetylated gangliosides and impaired the generation of 7,9-di-*O*-acetylated sialoglycans. Together, these findings establish a prominent role of SLC33A1 in the Golgi and demonstrate that the importance of the transporter extends beyond its relevance to protein acetylation in the ER[27]. Future work will require validation in more physiologically relevant models, such as primary cells or animal systems, as the current study relies on cell lines and protein overexpression.

Animal models highlighted that SLC33A1 is essential for mammalian development[24,60,61]. Defining the full SLC33A1 acetylation network remains key to understand the underlying mechanism and, moreover, to gain insight into the pathology of a spectrum of rare genetic disorders associated with mutations in *SLC33A1*, ranging from late-onset cerebellar ATX to SPG42 and childhood-onset fatal HBS[30,32,33]. Our functional analysis of four patient-derived SLC33A1 variants uncovered their markedly reduced ability to restore the formation of 9-*O*-Ac-GD3 in *Slc33a1* KO cells. Notably, the two HBS-associated variants almost failed to restore ganglioside 9-*O*-acetylation and it is likely that patients carrying loss-of-function mutations in *SLC33A1* show impaired ganglioside 9-*O*-acetylation. Deficient Sia *O*-acetylation may contribute to the neuropathological manifestation, given that 9-*O*-acetylated forms of complex b-series gangliosides such

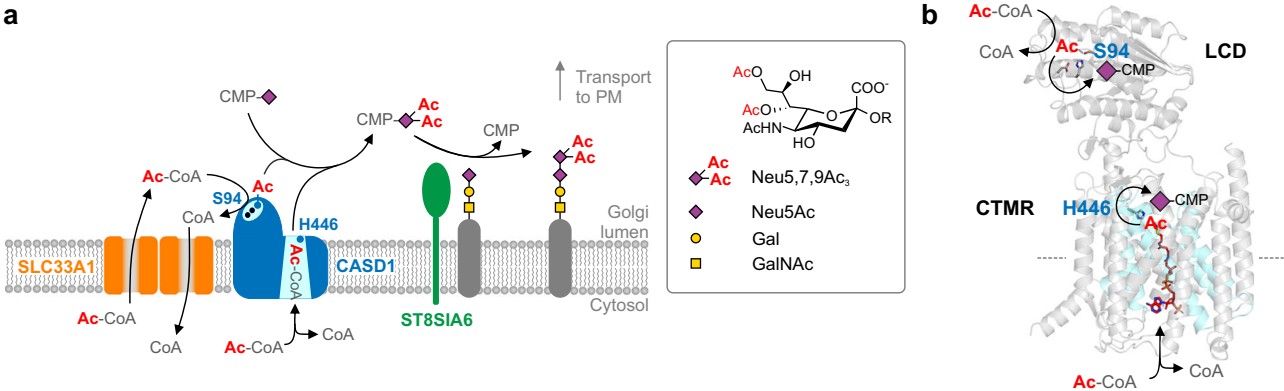

**Fig. 7 | Two-catalytic sites model for CASD1. a** Cartoon model for the formation of 7,9-*O*-acetylated sialosides depicting the concerted action of the acetyl-CoA transporter SLC33A1, the two catalytic domains of CASD1 and the sialyltransferase ST8SIA6. The two catalytic sites of CASD1 are shown in cyan with the key catalytic residues S94 and H446 depicted as blue dots. **b** AF2 model of CASD1 in cartoon representation with the catalytic triad residues S94-H273-D270 of the LCD and the catalytic residue H446 of the CTMR shown in stick representation. The transfer of

acetyl groups from luminal and from membrane-bound acetyl-CoA to CMP-Sia is depicted schematically for the LCD and the CTMR, respectively. TMHs forming the catalytic core of the CTMR are colored in cyan. The acetyl-CoA molecule shown within the catalytic core (red; stick representation) was taken from the superposition with HGSNAT in the acetyl-CoA bound state shown in Fig. 6c. Ac-CoA acetyl-coenzyme A, CTMR C-terminal transmembrane domain, LCD luminal catalytic domain, PM plasma membrane.

as GD1b and GT1b have neuroprotective effects[19] and that mutations in B4GALNT1, the glycosyltransferase required for the synthesis of complex b-series gangliosides, causes spastic paraplegia with cognitive impairment and developmental delay[62].

By unraveling the Golgi-specific function of SLC33A1 and the role of the CTMR of CASD1 in TM acetylation, we have advanced our mechanistic understanding of the Sia O-acetylation machinery and provided insight into how acetyl units are delivered to the Golgi lumen. As our knowledge on the entire acetylation landscape in the secretory pathway is still evolving, we hope that our study stimulates future research to explore the precise role of ER and Golgi acetylation in development and disease.

## Methods

All plasmids generated in this study, together with the sequences of the primers used for their construction, are provided in Supplementary Data 1. This Excel file contains multiple worksheets listing the primers used for the construction of SLC33A1, sialyltransferase and CASD1 plasmids, as well as the oligonucleotides employed for the insertion of guide RNAs into CRISPR/Cas plasmids. All primers and oligonucleotides were synthesized by Sigma-Aldrich. Newly generated plasmids described in this study are available from the corresponding author upon reasonable request.

### Generation of expression plasmids

For the generation of the plasmid pcDNA3-Flag-SLC33A1-Myc, which encodes human SLC33A1 with an N-terminal Flag- and a C-terminal Myc-tag, we first modified pcDNA3 (Invitrogen) by inserting Flag- and Myc-tag coding sequences into the HindIII/KpnI and XhoI/XbaI restriction sites, respectively, through adapter ligation of the oligonucleotide pairs FLAG-HK-F/FLAG-HK-R and MYC-F/MYC-R. The coding region of human *SLC33A1* (accession no. NM_004733.4) was amplified by RT-PCR from total RNA of SK-MEL-28 cells (CLS Cell Lines Service) using the gene-specific primers MG15 and MG16. The resulting PCR product was then ligated into the KpnI/XhoI sites of the modified pcDNA3 vector. Mutations underlying the SLC33A1 variants p.A110P, p.Y366*, p.S113R and p.G509S were introduced into pcDNA3-Flag-SLC33A1-Myc by fusion PCR using the primers MG15 and MG16 in combination with the following mutagenesis primer pairs: MG17/MG18 for generating pcDNA3-Flag-SLC33A1(A110P)-Myc, MG21/MG22 for generating pcDNA3-Flag-SLC33A1(Y366*), MG19/MG20 for generating pcDNA3-Flag-SLC33A1(S113R)-Myc, and MG23/MG24 for generating pcDNA3-Flag-SLC33A1(G509S)-Myc. The plasmid pcDNA3-V5-SLC33A1-Myc was generated from pcDNA3-Flag-SLC33A1-Myc by replacing the Flag-tag coding HindIII/KpnI-fragment by the V5-tag encoding oligonucleotide pair MH178/MH179.

For the generation of pcDNA3-Myc-CASD1, the coding region of human CASD1 was amplified by PCR using the primer pair MaA146/MaA147 and pcDNA3-V5-CASD1-Myc[20] as template. The resulting PCR product was ligated into the BamHI/XhoI sites of pcDNA3 (Invitrogen). Prior to this ligation step, the Myc-tag coding sequence was inserted into the HindIII/BamHI sites using the oligonucleotides Myc7s and Myc7as. The plasmid pcDNA3-Myc-CASD1(S94A) was generated from pcDNA3-Myc-CASD1 by replacing the BamHI/EcoRI fragment by the corresponding fragment excised from pcDNA3-V5-CASD1(S94A)-Myc[20]. The plasmid pcDNA3-Myc-V5-CASD1 was generated by insertion of the V5-tag encoding oligonucleotides MH258/MH259 into the BamHI site of pcDNA3-Myc-CASD1. The plasmid pcDNA3-Myc-V5-CASD1(S94A) was generated from pcDNA3-Myc-V5-CASD1 by replacing the SacII/EcoRI fragment by the corresponding fragment excised from pcDNA3-Myc-CASD1(S94A). The plasmid pcDNA3-Myc-V5-CASD1(S94A;H446A) was generated by fusion PCR using pcDNA3-Myc-V5-CASD1(S94A) as template and the mutagenesis primer pair LyB8/LyB7 in combination with the flanking primers LyB6 and AMB46. The resulting PCR fragment was ligated into the SacII/XbaI sites of pcDNA3-

Myc-V5-CASD1, thereby replacing the original SacII/XbaI fragment. All other amino acid exchanges within the CTMR of CASD1 were introduced accordingly using either pcDNA3-Myc-V5-CASD1 or pcDNA3-Myc-CASD1(S94A) as template and the mutagenesis primers listed in Supplementary Data 1. For the generation of the plasmid pcDNA3-Myc-V5-CASD1-ΔLCD, the sequence encoding the LCD of CASD1 (amino acid residues 61-300) was substituted by an in-frame insertion of a sequence stretch that encodes the FLAG epitope MDYKDDDD. In a first step, the sequence stretches flanking the deletion site were amplified with the primers LyB6/AMB47 and AMB48/AMB46, of which AMB47 and AMB46 contain complementary sequence stretches that encode the FLAG epitope. In a second step, the obtained amplification products were fused in a PCR reaction with the primer pair LyB6/AMB46, and the final product was ligated into the SacII/XbaI sites of pcDNA3-Myc-V5-CASD1, replacing the original SacII/XbaI fragment.

The plasmid pcDNA3-V5-ST8SIA1 was generated by cloning the HindIII/NotI fragment excised from pcDNA3.1-zeo-V5-ST8SIA1[20] into the HindIII/NotI sites of the pcDNA3 (Invitrogen). The bi-cistronic construct pcDNA3-ST8SIA1-2A-B4GALNT1 carries the sequence encoding human ST8SIA1 (NM_003034.3) without stop-codon followed in-frame by the sequence encoding the 2A "self-cleaving" peptide from equine rhinitis A virus (QCTNYALLKLAGDVESNPGP) and the sequence encoding human B4GALNT1 (NM_001478.5). This tripartite coding sequence was generated by PCR in three steps: (i) amplification of a fragment encoding ST8SIA1 fused to the 2A peptide using the primer pair LS58/LS67 and pcDNA3-V5-ST8SIA1 as template, (ii) amplification of a fragment encoding the 2A peptide fused to B4GALNT1 using the primer pair LS66/LS62 and pTWIST-Amp-High-Copy-hB4GALNT1 as template (Twist Biosciences, insert generated by custom DNA synthesis), and (iii) fusion PCR with the flanking primers LS58 and LS62. The resulting PCR fragment was ligated into the NotI/XbaI sites of pcDNA3. The plasmid pcDNA3-V5-ST8SIA6 was generated by the amplification of the coding sequence of human *ST8SIA6* (accession no. NM_001004470.2) using the primer pair MaA90/MaA93 and pcDNA3.1*/C-(K)-DYK-ST8SIA6 (GeneScript, cat. OHu03577) as template. The resulting PCR product was ligated into the BamHI/XbaI restriction sites of pcDNA3-V5-CASD1-Myc, thereby replacing the original CASD1-Myc encoding BamHI/XbaI fragment by the ST8SIA6 encoding sequence. The identity of all plasmids was confirmed by Sanger sequencing. Sequences of the PCR primers are listed in Supplementary Data 1.

### Cell culture

HAP1 cells (Haplogen Genomics, Austria; RRID: CVCL_Y019) were grown in Iscove's Modified Dulbecco's Medium (Gibco) supplemented with 5% fetal calf serum (FCS) and 1 mM sodium pyruvate. Chinese Hamster Ovary K1 (CHO-K1) cells were cultured in Dulbecco's Modified Eagle's Medium (DMEM)/Ham's F12 1:1 (PAN-Biotech) supplemented with 5 % FCS. Cultures were maintained at 37 °C with a humidified atmosphere of 5% $CO_2$.

### CRISPR/Cas9-mediated genome editing

HAP1-Δ*SLC33A1* cells (HAP1_SLC33A1_36819-02) were custom-made by Haplogen Genomics (Austria), using the *Streptococcus pyogenes* nuclease Cas9 (SpCas9) together with a guide RNA containing the gene-specific sequence 5′-GTAGACCGCATCAACCAACG-3′ for targeting exon 1 of human *SLC33A1*. The obtained HAP1-Δ*SLC33A1* clone carries a 2-bp frameshift deletion in exon 1 of *SLC33A1* as shown in Supplementary Fig. 1.

CHO-Δ*Casd1* cells were established previously[21] by introducing a frameshift mutation in exon 2 of *Casd1* using the single guide RNA strategy described below. CHO-Δ*Slc33a1* cells were generated by using the pX330-U6-Chimeric_BB-CBh-hSpCas9 plasmid (Addgene plasmid # 42230, a gift from Feng Zhang)[63] with the gene-specific target sequence inserted into the BsbI sites of the guide RNA scaffold.

CCTop-CRISPR/Cas9 target online predictor[64] was used for the identification of target sites. Information on the selected sites and the respective oligonucleotide sequences used for the generation of single guide RNAs targeting exon (Ex) 1, 3 or 5 of *Slc33a1* are provided in Supplementary Data 1. The resulting plasmids pX330-*Slc33a1*-Ex1, pX330-*Slc33a1*-Ex3, and pX330-*Slc33a1*-Ex5 were introduced into CHO cells, either individually for single guide RNA approaches to induce frameshift mutations or as a combination of pX330-*Slc33a1*-Ex1 and pX330-*Slc33a1*-Ex5 to introduce a multiple exon-spanning deletion. To facilitate the selection of transfected cells, an enhanced green fluorescent protein (EGFP) reporter plasmid (pEGFP-C1, Clontech) was co-transfected in a 4:1 ratio of pX330-based plasmid(s) to pEGFP-C1. Single green fluorescent colonies were selected and genotyped by PCR and Sanger sequencing. Clones with homozygous or heterozygous frameshift mutations or multiple-exon spanning deletions were subcloned and genotyped on genomic and transcript level. For the analysis of genomic DNA, $2 \times 10^5$ cells were lysed in 20 μL of lysis buffer (10 mM Tris-HCl pH7.6, 50 mM NaCl, 6.25 mM MgCl₂, 0.045 % NP-40, 0.45 % Tween-20) containing 1 mg mL⁻¹ proteinase K. After an incubation for 1 h at 56 °C and heat-inactivation of the proteinase for 15 min at 95 °C, an aliquot of 1 μL of lysate was analyzed by PCR using the following primer combinations: 5′-CCGGGGATCTGGCTCTGTGC-3′ and 5′-GGATGCTCCCTGCCAGGCC-3′ for amplification of the Ex1 target site, 5′-CAGTTCTACCAATATTGCAT-3′ and 5′-GTGCACAATACTGAACACA-3′ for amplification of the Ex3 target site, 5′-TTCCACCATTCCAGTGTTCTCT-3′ and 5′-ATTTCTAAGCAGTTCAGCAACTCTC-3′ for amplification of the Ex5 target site, and 5′-ATGTCACCGACCATCTCC-3′ and 5′-ATTTCTAAGCAGTTCAGCAACTCTC-3′ for the analysis of deletions spanning exon 1 to 5. The first primer of each combination was also used for Sanger sequencing of the obtained PCR products. Mutations were confirmed on transcript level by Sanger sequencing of RT-PCR products, which were obtained from total RNA using the following primer pairs: 5′-ATGTCACCGACCATCTCC-3′ (5′-end of exon 1) and 5′-GAGGGACCATTGGAACAGCA-3′ (exon 3), 5′-AACGTGGGC-TATGCTTCCAC-3′ (3′-end of exon 1) and 5′-AGAGCTGTCACA-CATGAGCC-3′ (exon 6), and 5′-ATGTCACCGACCATCTCC-3′ (5′-end of exon 1) and 5′-AGAGCTGTCACACATGAGCC-3′ (exon 6).

## Transfection of mammalian cells
If not stated otherwise, transfections were performed with Polyethylenimine 'Max' (PEI MAX, Polysciences). Cells were grown to 70–80 % confluency in 6-well plates containing 2 mL of culture medium per well or, alternatively, on 10 cm dishes containing 10 mL of culture medium. Per well of a 6-well plate, 2 μL PEI MAX were diluted in 100 μL of OptiMEM (Gibco) and mixed with 100 μL of OptiMEM containing 2 μg of plasmid DNA. After an incubation for 15–20 min at room temperature, the PEI MAX-DNA mixture was added to the culture medium. Cells grown in 10 cm dishes were transfected accordingly with reagent amounts scaled by a multiplication factor of 5. Transfections were stopped after 6 h by replacing the PEI MAX-containing medium by fresh culture medium. Cells were harvested 48 h after transfection for ganglioside analyses and after 24 h for immunofluorescence microscopy. For the subcellular localization analysis shown in Fig. 2h, transfections were performed with Metafectene (Biontex) instead of PEI MAX. All steps were performed in 6-well plates as described above, except that 3 μL of Metafectene and 0.5 μg of plasmid DNA were used per well.

## Immunofluorescence analysis
Cells were seeded on glass coverslips placed in 6-well plates and transfected as described above. Adherent cells were washed with PBS, fixed with 4 % paraformaldehyde in PBS for 15 min at room temperature, and washed again with PBS. For the detection of Golgi localized proteins, cells were permeabilized with 0.2 % (v/v) Triton X-100 in PBS for 10 min at 4 °C. After blocking for 30 min with 1 % bovine serum

albumin (BSA) in PBS, cells were stained for 1 h at room temperature with the following primary antibodies or virolectins diluted in 1% BSA in PBS: mouse anti-9-*O*-Ac-GD3 mAb (1 μg mL⁻¹; Santa Cruz cat. Sc-32269, clone UM4D4), mouse anti-GD3 mAb R24 (2.5 μg mL⁻¹; purified by protein A affinity chromatography from cell culture supernatant of hybridoma cells ATCC HB-8445), mouse anti-Flag mAb (5 μg mL⁻¹; Sigma-Aldrich cat. F1804, clone M2), rabbit anti-Flag pAb (DYKDDDDK-Tag antibody; 0.28 μg mL⁻¹; Cell Signaling cat. 2368), mouse anti-Myc mAb (10 μg mL⁻¹; ThermoFisher, MA1-980, clone 9E10), mouse anti-V5 mAb (1 μg mL⁻¹; Acris cat. SM1691PS, clone SV5-PK1), chicken anti-V5 pAb (2 μg mL⁻¹; Abcam cat. ab9113), rabbit anti-α-Man II pAb (1:10,000; kindly provided by Kelley Moremen), BCoV-HE⁰-Fc (40 μg mL⁻¹), and ICV-HE⁰-Fc (10 μg mL⁻¹). Cells were washed and incubated with secondary antibody (diluted in 1 % BSA in PBS) for 1 h at room temperature: goat anti-chicken IgY Alexa Fluor 488-conjugate (1:1000; Invitrogen cat. A-11039), donkey anti-human IgG Dylight 550-conjugate (1:1000; Invitrogen cat. SA5-10127), goat anti-mouse IgG Alexa Fluor 488-conjugate (1:500; Invitrogen cat. A-11029), sheep anti-mouse IgG Cy3-conjugate (1:1000; Sigma-Aldrich cat. C2181), donkey anti-mouse IgG Alexa 555 (1:500; Invitrogen cat. A-31570), goat anti-mouse IgG2a Alexa Fluor 488-conjugate (1:500; Invitrogen cat. A-21131) rabbit anti-mouse IgG3 DyLight 549-conjugate (1:5000; Rockland cat. 610-442-043), goat anti-mouse IgM Alexa Fluor 568-conjugate (1:500; Invitrogen cat. A-21043), goat anti-rabbit IgG Alexa Fluor 488-conjugate (1:500; Invitrogen cat. A-11008), sheep anti-rabbit IgG Cy3-conjugate (1:1000; Sigma-Aldrich cat. C2306). After washing with PBS and H₂O, air-dried cells on cover slips were mounted in 3 μL Vectashield mounting medium containing 4′,6-diamidino-2-phenylindole (DAPI) (Vector Laboratories). Stained cells were analyzed using an Axiovert 200 M microscope equipped with an ApoTome module and an Axio-Cam MRm digital camera (Zeiss). Images were processed using the software ZEN 2012 (Zeiss).

## Glycan profiling
Profiling of GSL-derived glycans by multiplexed capillary gel electrophoresis coupled to laser-induced fluorescence detection (xCGE-LIF) was performed as described[65]. Briefly, total GSLs extracted from 10⁷ cells were digested with 1 μL of LudgerZyme Ceramide Glycanase (Ludger) in a total volume of 6.5 μL LudgerZyme Ceramide Glycanase RXN Buffer for 24 h at 37 °C. Samples were dried and mixed with 2 μL of 8-aminopyrene-1,3,6-trisulfonic acid trisodium salt (APTS, Sigma-Aldrich; 20 nM in 3.5 M citric acid, 2 μL of 2-picoline borane complex (Sigma-Aldrich; 2 M in DMSO) and 2 μL of H₂O. The reaction mixture was incubated at 37 °C in the dark and stopped after 16.5 h by the addition of 100 μL of acetonitrile/H₂O (80:20, v/v). APTS-labeled glycans were purified by hydrophilic interaction liquid chromatography-solid phase extraction (HILIC-SPE) and analyzed by xCGE-LIF using a remodeled ABI PRISM 3100-Avant genetic Analyzer (Applied Biosystems). Peaks were assigned according to the migration time of structurally defined glycan standards.

## Ganglioside extraction and immuno-thin-layer chromatography
Total gangliosides were extracted with chloroform/methanol (1:2, v/v; 3 mL per 10⁷ cells) in conjunction with ultrasonic dispersion, followed by solid-phase extraction on Chromabond C18 columns (Macherey-Nagel)[21,39]. For TLC, total gangliosides of an equivalent of $2 \times 10^6$ cells were spotted on Nano-DURASIL-20 HPTLC plates coated with 0.2 mm silica gel 60 (Macherey-Nagel). Plates were chromatographed in chloroform/methanol/H₂O (50:40:10, v/v/v) containing 0.05% calcium chloride and dried. Plates were then chromatographed twice in 0.5% isobutyl methacrylate polymer (TCI, cat. M0086), dried and incubated overnight in PBS at 37 °C. After blocking with 2% BSA (w/v) in PBS for 1 h at room temperature, plates were washed with PBS and stained for 1 h at room temperature with the following primary antibodies diluted in PBS: mouse IgM anti-9-*O*-

Ac-GD3 mAb (5 µg mL$^{-1}$; Ancell, ANC-212-820, clone UM4D4), mouse IgM anti-9-O-Ac-GD3 mAb (1:40; Thermo Scientific; cat. MA1-34707, clone M-T6004), mouse IgG3 anti-GD3 mAb R24 (10 µg mL$^{-1}$; affinity purified from cell culture supernatant of hybridoma cells ATCC HB-8445)[20], mouse IgG3 anti-9-O-Ac-GD2 mAb 8B6 (10 µg mL$^{-1}$; kindly provided by OGD2 Pharma, Nantes), or mouse IgG2a anti-GD2 mAb (15 µg mL$^{-1}$; Kerafast EWI023, clone ME361). Plates were washed with PBS and incubated with the following secondary antibodies for 1 h at room temperature: goat anti-mouse IgG IRDye 800CW-conjugate (1:10,000; LI-COR Biosciences, cat. 926-32210) or goat anti-mouse IgM IRDye 800CW-conjugate (1:20,000; LI-COR Biosciences, cat. 926-32280). TLC plates were washed and signals were recorded using an infrared imaging system (Odyssey Imaging System, LI-COR Biosciences). Purified GD3 (Sigma-Aldrich, cat. 345752) and GD2 (Sigma-Aldrich, cat. 345743) were used as standards (0.2 µg per lane). The corresponding 9-O-acetylated forms were generated by in vitro O-acetylation of purified GD2 and GD3 using the *Campylobacter jejuni* O-acetyltransferase NeuD, recombinantly expressed as maltose-binding protein (MBP) fusion protein in *Escherichia coli*[21,39]. In a total volume of 20 µl, 27.5 µM ganglioside were incubated in the presence of 10 µg MBP-NeuD, 1.5 mM acetyl-CoA (Sigma-Aldrich), 0.05% sodium cholate, 10 mM MgCl$_2$, 1 mM dithiothreitol and 50 mM MES buffer pH 7.2. After 6 h at 37 °C, the reaction was stopped by adding an equal volume of methanol. Gangliosides were purified by solid-phase extraction on Chromabond C18 columns (Macherey-Nagel) and dissolved in chloroform/methanol (1:2, v/v). Uncropped images of the TLC plates are available in the Source Data file.

## Cell viability assay
Cell viability and proliferation was measured in a tetrazolium dye assay using water-soluble tetrazolium reagent-1 (WST-1) according to the manufacturer's protocol (Roche). CHO cells were seeded at a density of 500 cells per 100 µL culture medium per well of a 96-well culture plate and cultivated for 24 h, 48 h, 72 h or 96 h. The medium was changed, followed by the addition of 10 µL of WST-1 assay reagent. After 2 h of incubation at 37 °C and 5% CO$_2$, absorbance at 450 nm was measured against a reference wavelength at 690 nm using a microplate reader (PowerWave 340, Bio-Tek). Cell-free wells were used as blank control.

## Acetyl-CoA measurements
CHO cells were seeded at a density of $3 \times 10^6$ cells per 10 cm dish and allowed to adhere for 24 h at 37 °C and 5% CO$_2$. Acetyl-CoA was extracted from cells by chloroform-methanol liquid-liquid extraction[66]. Adherent cells were washed three times with ice-cold PBS and cellular metabolism was quenched by the addition of 500 µL of methanol pre-cooled to −20 °C. Isotopically U-$^{13}$C-labeled metabolite yeast extract (Isotopic solutions, cat. G83-14-07-2021) was dissolved in water and used as internal standard. An aliquot corresponding to 150 µg of yeast dry cell weight in a total volume of 1 mL dest. H$_2$O was added to each dish and cells were scraped and transferred to a microfuge tube containing 500 µL of chloroform pre-cooled to -20 °C. Cell extraction was performed for 20 min at 4 °C in an Eppendorf mixer at 1400 rpm. After phase separation by centrifugation (5 min at $16,100 \times g$, 4 °C), the interphase was retained for protein extraction as described below and acetyl-CoA was isolated from the polar phase by 2-(2-pyridyl)ethyl silica gel based solid-phase extraction (SPE)[67]. SPE columns packed with functionalized (2-pyridyl)ethyl silica gel (100 mg, Merck, cat. 54127-U) were equilibrated with 1 mL of equilibration buffer (45% ACN, 20% H$_2$O, 20% acetic acid, 15% isopropanol (v/v), pH 3). After equilibration, the samples (dissolved in the polar methanol phase) were loaded on the SPE column and washed with 1 mL of the equilibration buffer. Acetyl-CoA was eluted from the column using 2 mL of methanol/250 mM ammonium formate 4:1 (v/v, pH 7). The solvent was evaporated under a nitrogen stream while samples were kept on ice

and dried samples were stored at −80 °C. Prior to analysis, samples were reconstituted with 100 µL 50% ACN in dest. H$_2$O. Acetyl-CoA levels were quantified by hydrophilic interaction chromatography (HILIC) coupled to mass spectrometry (HILIC-MS)[68]. An aliquot of 10 µL was injected into an UPLC system (Vanquish Flex, Thermo Scientific) equipped with a zwitterionic HILIC column (Atlantis Premier BEH z-HILIC VanGuard FIT, 1.7 µm, 2.1 × 150 mm, Waters). Analytes were loaded for 1 min at 90% B, and separated using a linear gradient ranging from 90 to 65% B in 5 min with a flow rate of 500 µL min$^{-1}$ (A: 15 mM ammonium bicarbonate (AmBiCa) in H$_2$O, pH 9; B: 90% ACN, 15 mM AmBiCa in H$_2$O, pH 9). The UPLC system was coupled via an electrospray-ionization (ESI) source to a tribrid orbitrap mass spectrometer (Orbitrap Fusion Lumos, Thermo Scientific). Single ion monitoring (SIM)-MS analysis of acetyl-CoA was performed in positive ion mode to quantify acetyl-CoA. Parameters were set as follows: orbitrap resolution: 60k (at m/z 200), maximum ion injection time: 118 ms, normalized AGC target: 100%, RF lens: 50%, isolation windows: $^{12}$C-acetyl-CoA: 810.1330 ± 5 m/z, U-$^{13}$C-acetyl-CoA: 833.2102 ± 5 m/z, retention time: 2.8–3.8 min. Samples were measured in a randomized sequence. An external standard row with 7 standards ranging from 0.19 ng on column to 12.5 ng was measured before and after the samples to enable quantification.

LC-MS raw data were processed using TraceFinder (version 5.0.889.0, Thermo Fisher Scientific) with the following parameters: detection algorithm: Genesis, peak detection strategy: highest peak, peak area: 1%, smoothing: 5, min peak height S/N: 2. Manual integration of peaks was performed when necessary. Acetyl-CoA data from cell extracts were normalized to U-$^{13}$C-labeled acetyl-CoA as internal standard to compensate for any possible variations during sample processing and to total protein content as a surrogate for cell number. For protein determination, the interphase of each cell extract was washed with methanol, pelleted by centrifugation (10 min at $16,100 \times g$, 4 °C) and air dried at 35 °C for 15 min. Proteins were dissolved in 500 µL of 8 M urea, assisted by sonication (Branson Ultrasonic Sonifier S-450 equipped with a cup horn). The bicinchoninic acid (BCA) protein assay was performed on serially diluted samples according to the manufacturer's protocol (Pierce).

## Western Blot analysis of whole cell lysates
For the detection of Flag-tagged SLC33A1 variants, cell lysates were prepared using n-dodecyl-β-D-maltopyranoside (DDM; Anatrace) as detergent. A total number of $1 \times 10^6$ transfected CHO cells were lysed in 70 µL of DDM-lysis buffer (50 mM Tris-HCl pH 8.0, 150 mM NaCl, 5 mM EDTA, 1% DDM, 200 U mL$^{-1}$ aprotinin, 10 µg mL$^{-1}$ leupeptin, 2 mM PMSF) for 15 min on ice. After the removal of cell debris by centrifugation at $13,000 \times g$ for 15 min at 4 °C, lysates were mixed with an equal volume of 2-fold concentrated Laemmli sample buffer. After an incubation for 10 min at 65 °C, samples of an equivalent cell number ($9 \times 10^4$ cells per lane) were separated by 10 % SDS-PAGE, and proteins were transferred to polyvinylidene difluoride (PVDF) membrane by semi-dry blotting. Membranes were blocked in 3% BSA in PBS and incubated with rabbit anti-Flag pAb (DYKDDDDK-Tag antibody; 0.02 µg mL$^{-1}$; Cell Signaling cat. 2368) overnight at 4 °C. Membranes were washed and incubated for 1 h at room temperature with goat anti-rabbit IgG horseradish peroxidase (POD)-conjugate (1:20,000; Sigma-Aldrich cat. A6154), followed by enhanced chemiluminescence (ECL) detection using an Amersham Imager 680 (Cytiva). Bound antibodies were stripped by incubating the membranes in stripping buffer containing 2% SDS and 0.8 % β-mercaptoethanol in 62 mM Tris-HCl (pH 6.8) for 30 min at 50 °C. After washing with H$_2$O, membranes were blocked and stained with mouse anti-actin mAb (1:100,000; Merck Millipore cat. MAB1501, clone C4) and goat anti-mouse IgG POD-conjugate (1:20,000; SouthernBiotech cat. 1010-05) following the same protocol as described above. All antibodies were diluted in blocking solution (3% BSA in PBS).

For the detection of V5-tagged ST8SIA6 (Supplementary Fig. 11), $1 \times 10^6$ cells were lysed in 70 μL of Triton-lysis buffer (50 mM Tris-HCl pH 8.0, 150 mM NaCl, 5 mM EDTA, 2 % Triton X-100 (v/v), 0.5 % sodium deoxycholate (w/v), 200 U mL$^{-1}$ aprotinin, 10 μg mL$^{-1}$ leupeptin, 2 mM PMSF). Samples were processed as described above and an equivalent of $7 \times 10^4$ cells was applied per lane. After blotting, PVDF membranes were blocked with Odyssey blocking buffer (LI-COR Biosciences) diluted 1:2 with PBS. Antibody staining was performed in the same buffer system using mouse anti-V5 mAb (0.1 μg mL$^{-1}$; Acris cat. SM1691PS, clone SV5-PK1) followed by goat anti-mouse IgG IRDye 800CW-conjugate (1:20,000; LI-COR Biosciences cat. 926-32210) and rabbit anti-Actin pAb (1:1000; Sigma-Aldrich cat. A2066) followed by anti-rabbit IgG IRDye 800CW-conjugate (1:20,000; LI-COR Biosciences cat. 926-32211). Signals were detected by an infrared imaging system (Odyssey Imaging System, LI-COR Biosciences). Uncropped images of the blots are available in the Source Data file.

### Structure prediction and analyses

The Protein Data Bank (PDB) coordinate files of the AF2 models of human CASD1 and Salmonella OafB were downloaded from the AlphaFold Protein Structure Database[69,70] with the accession AF-Q96PB1-F1 and AF-A0A0H2WM30-F1, respectively. The PyMOL Molecular Graphics System (DeLano Scientific) was used for visual inspection of the structural models and for producing graphical images. Superpositions of structures were obtained by employing the super command implemented in PyMOL.

### Expression and purification of virolectins

The virolectins BCoV-HE$^0$-Fc and ICV-HE$^0$-Fc were produced as secreted soluble Fc-chimera in baculoviral infected *Trichoplusia ni* (High 5) insect cells and were purified from the cell culture supernatant by protein A affinity chromatography[39]. The expressed proteins encompass an enzymatically inactive HE ectodomain (amino acid residues 19-388 of BCoV-LUN HE carrying an S40A exchange or residues 15-631 of influenza C/California/78 HE carrying an S71A exchange) followed by the Fc part of human IgG1[20,39].

### Flow cytometry

Cells were washed with excess of FACS buffer (PBS containing 1% BSA and 1 mM EDTA) and adjusted to $2.5 \times 10^5$ cells per 100 μL of FACS buffer. Samples were stained for 1 h at 4 °C with the following primary antibodies or virolectins in 100 μL FACS buffer: anti-9-*O*-Ac-GD3 mAb (1:20; mouse IgM, Thermo Scientific, cat. MA1-34707, clone M-T6004), BCoV-HE$^0$-Fc (40 μg mL$^{-1}$), or ICV-HE$^0$-Fc (10 μg mL$^{-1}$). Cells were washed twice and incubated with secondary antibody for 30 min at 4 °C using rat anti-mouse IgM PE-conjugate (1:80; BD cat. 553409) or goat anti-human IgG R-Phycoerythrin (R-PE)-conjugate (0.5 μg mL$^{-1}$; Jackson ImmunoResearch, cat. 109-115-098). Cells were washed and suspended in 1 mL of FACS buffer. 15,000 events per sample were measured on a CyFlow ML flow cytometer (Sysmex Partec) equipped with a 488 nm laser. Data were analyzed using the FlowJo software Version 7.6.

### Enzymatic de-*O*-acetylation

To validate specific virolectin staining, *O*-acetyl groups were removed by the sialate-7,9-*O*-acetylesterase NanS from *Tannerella forsythia*. Recombinant NanS carrying a C-terminal hexa-histidine-tag was expressed in *Escherichia coli* and purified by affinity chromatography[39]. Cells were adjusted to $2.5 \times 10^5$ cells per 100 μL of FACS buffer and incubated in the presence of 5 μg of NanS for 1 h at room temperature. Afterwards, cells were stained and analyzed by flow cytometry as described above.

### Computational methods

All molecular dynamics (MD) simulations were performed using the GPU version of Gromacs 2022[71,72]. The Gromos 54A7[73,74] force field was used to model protein as well as the model lipid bilayer (DPPC: 1,2-dipalmitoyl-sn-*glycero*-3-phosphocholine) membrane[75,76]. The force field parameters for the ligand CMP-Neu5Ac were obtained from the GROLIGFF[77] webserver. A model structure of the human CASD1 TM domain spanning P307-H796 was generated by AlphaFold2 using the ColabFold platform with default parameters[69,78]. The CMP-Neu5Ac ligand was positioned in the binding pocket by using structure-guided manual docking based on the co-crystal structures of Heparan α-glucosamine *N*-acetyltransferase (PDB: 8JKV, 8JL1, 8JL3, 8JL4, 8TU9, 8VKJ, 8VLG, 8VLI, 8VLU, 8VLY). The protonation state of titratable groups was chosen appropriate to pH 7.0. Each ligand-protein complex was embedded into a model membrane, composed of 512 DPPC molecules, using the GROMACS program g_embed. The complex was resized in the XY directions (corresponding to the plane of the membrane) to 10% of their original size and to insert the α-helices any overlapping lipid and water molecules were deleted. The complex was then grown stepwise to the original size. The final simulations rectangular box had approximate dimensions of 12.9 nm × 12.9 nm × 8.9 nm and contained ~27,000 water molecules and overall ~111,000 atoms (see Supplementary Fig. 18a).

Each system was simulated under periodic boundary conditions in a rectangular box. The pressure was maintained at 1 bar by weakly coupling the system to a semi-isotropic pressure bath using an isothermal compressibility of $4.6 \times 10^{-5}$ bar$^{-1}$ and a coupling constant $\tau_P = 1$ ps in the XY and Z directions, corresponding to the plane and normal directions of the bilayer, respectively[79]. The temperature of the system was maintained at 298 K by independently coupling the protein-ligand complex, lipids and water to an external temperature bath with a coupling constant $\tau T = 0.1$ ps using a Berendsen thermostat[79]. A nonbonded interaction cut-off of 0.14 nm was used. Long-range electrostatics were treated with the Reaction Field[80] method. All bond lengths within the protein and lipids were constrained using the LINCS[81] algorithm. The simple-point charge (SPC)[82] water model was used and constrained using the SETTLE[83] algorithm. Each system was energy minimized for 1000 steps using steepest descent method followed by a position-restrained MD simulation where all heavy atoms of protein-ligand complex were restrained to their original position using 1000 kJ/mol/nm$^2$ allowing lipid and water molecules to equilibrate. The restraints were removed and the whole system was allowed to equilibrate for 5 ns. The production MD simulations were performed for 2000 ns (in triplicate–two starting with a different initial velocity distribution and one starting from a different configuration) for each system. Supplementary Fig. 18 shows that substrate relaxation and binding orientation are reached within the simulated timeframe, consistent with the process being accessible on the brute-force MD timescale used here. All coordinates, velocities, forces, and energies were saved every 10,000 steps for analysis. The stability of the protein and protein-ligand complexes were evaluated by measuring the root mean square deviation (RMSD) of protein backbone as well as ligand atoms by fitting the backbone atoms of protein. The molecular dynamics checklist can be found in Supplementary Table 3. The molecular dynamics simulation data (starting structure, force field parameter files, molecular dynamics parameter, and trajectories) generated in this study have been deposited in the Zenodo database (https://doi.org/10.5281/zenodo.18795789)[84].

### On-bead *O*-acetyltransferase assay

To measure CASD1-mediated SOAT activity, we performed an on-bead enzyme reaction with affinity-captured CASD1 as enzyme source, and quantified the acetyltransferase reaction product HS-CoA via fluorescence-based thiol detection. Briefly, we expressed WT and mutant forms of N-terminally Myc-V5-tagged CASD1 in CHO-Δ*Casd1* cells using the PEI transfection protocol described above. For each enzyme reaction, Myc-V5-CASD1 was affinity captured from the detergent lysate of $1.5 \times 10^7$ cells harvested 48 h after transfection.

Lysis was performed in 600 μL DDM-lysis buffer (50 mM Tris pH 8.0, 150 mM NaCl, 5 mM EDTA, 0.5% DDM, 200 U mL$^{-1}$ aprotinin, 10 μg mL$^{-1}$ leupeptin) for 1 h at 4 °C, and the obtained lysates were clarified by centrifugation at 16,100 × g for 15 min at 4 °C. Anti-Myc nanobody-coupled magnetic agarose beads (Myc-Trap, ChromoTek) were added to the cleared lysates using 40 μL of the Myc-Trap suspension, and the affinity capture was performed for 2 h at 4 °C on a rotary wheel. Beads were washed once with 1 mL of wash buffer (10 mM MES pH 6.5, 1 mM MnCl$_2$, 150 mM NaCl) with 0.05% DDM and twice without detergent, followed by two washes with reaction buffer (10 mM MES pH 6.5, 1 mM MnCl$_2$). Affinity beads incubated with the detergent lysate of mock-transfected cells were used as a non-enzyme control.

The on-bead enzyme reaction was performed in a total volume of 60 μL reaction buffer containing 1 mM acetyl-CoA trisodium salt (Sigma-Aldrich) and 1 mM CMP-Neu5Ac sodium salt (Roche) with the affinity-captured material as enzyme source. After a 3-h incubation at 37 °C with continuous rotary motion, beads were removed by magnetic separation. The bound protein was subsequently analyzed by Western blotting with an anti-V5 antibody as described above for the detection of V5-tagged ST8SIA6. The reaction mixture was passed through a 50 kD cut-off filter. Ten microliters of the ultrafiltrate were combined with 100 μL of Measure-iT Thiol Assay Reagent (Thermo Fisher Scientific) and the fluorescence intensity was measured using an Infinite 200 Pro microplate reader (Tecan) with $\lambda_{EX}/\lambda_{EM}$ = 485/520 nm. For the calculation of relative fluorescence intensities, the signal from the non-enzyme control was defined as 0% and the signal from the WT control was set to 100%. Statistical significance was determined by one-way ANOVA using GraphPad Prism v8.

### Reporting summary
Further information on research design is available in the Nature Portfolio Reporting Summary linked to this article.

## Data availability
The molecular dynamics simulations data (starting structure, force field parameter files, molecular dynamics parameter and trajectories) generated in this study have been deposited in the Zenodo database (https://doi.org/10.5281/zenodo.18795789)[84]. The source data underlying Figs. 2b, c, e, f, 3a, b, 4d, f, h, 5d, and 6f, g, and Supplementary Figs. 6, 11,and 19 are provided as a Source Data file. Source data are provided with this paper.

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

## Acknowledgements

This work was funded by the Deutsche Forschungsgemeinschaft (DFG, German Research Foundation) within the Research Unit 2953 (Project Nos. 432251600/FOR2953 to M.M. and 409784463/FOR2953 to M.M., F.F.R.B., and G.Z.). We acknowledge support from European Union's Horizon 2020 research and innovation program (MESI-STRAT Grant Agreement No. 754688 to K.T.), from the European Partnership for the Assessment of Risks from Chemicals PARC (Grant Agreement No. 101057014 to K.T.) and European Research Council (ERC AdG BEYOND STRESS, Grant Agreement No. 101054429 to K.T.) which have received funding from the European Union's Horizon Europe research and innovation program. We acknowledge the financial support of the National Health and Medical Research Council, Australia (NHMRC, ID 2009677 & GNT1196520 to M.v.I.) and Griffith University for the award of a Griffith University Postdoctoral Award (to T.L.). The computational work was funded with the assistance of high-performance computing resources provided through the National Computational Merit Allocation Scheme supported by the Australian Government (Project cj47) and Queensland Cyber Infrastructure Foundation (QCIF, Project fi49) and the Griffith University HPC facility. We thank Astrid Oberbeck for technical assistance and Rita Gerardy-Schahn for support and helpful discussions. Views & opinions are those of the authors.

## Author contributions

M.A., L.B., A.-M.T.J., L.S., C.R., M.H., and M.G. performed the experimental studies. T.L. and A.K.M. carried out the MD simulations. A.-S.E. and M.K. performed the LC-MS-based metabolite analysis, and A.-S.E., M.K., L.S., and K.T. analyzed the obtained data. G.Z. and M.M. performed structural modeling and analysis. F.F.R.B. analyzed the glycan profiling data. K.T., F.F.R.B, M.v.I., and M.M. supervised the work. M.A., L.B., and M.M. conceived and planned the experiments with input from M.v.I. M.A. and M.M. wrote the manuscript. All authors provided critical feedback and helped shape the research, analysis, and manuscript.

## Funding

## Competing interests

The authors declare no competing interests.
