## [Transparent Peer Review file · Nature Communications]

Interplay of SLC33A1-dependent and -independent Golgi sialic acid O-acetylation in CASD1 catalysis

Corresponding Author: Dr Martina Mühlenhoff

Version 0:

Reviewer comments:

Reviewer #1

(Remarks to the Author)

This manuscript by Albers et al. focuses on a membrane protein sialate O-acetyltransferase CAS1 domain-containing 1 (CASD1). The primary finding—that CASD1 possesses two active sites, one dependent on SLC33A1 (acetyl-CoA transporter) and one independent—is exciting and significantly advances our understanding of this enzyme and Golgi acetylation processes. The manuscript is well-structured, with a clear presentation of results, and this reviewer identifies no major concerns. I strongly support its publication in Nature Communications, with some minor concerns detailed below.

1. In Figure 2f, only the Ex 1-5 del clone is presented. Why were the other two Ex clones omitted? Including all three would allow readers to better assess clonal variation in the data. Additionally, could the authors clarify when acetyl-CoA levels were measured and whether these measurements align with the time points in Figure 2e?
2. In Figure 3b, the anti-Flag western blot shows an impurity band near the Flag-SLC33A1-Y336* truncated protein. Could the authors clarify its identity or source? The band's intensity appears correlated with ST8SIA1 expression.
3. The MD simulation details should be included in the main methods section rather than relegated to the supplementary material. Given the complexity of the simulation system, the energy minimization and equilibration steps seem brief. Furthermore, the production phase of only 100 ns is relatively short; with current computational resources, extending it to 1 μ s would be feasible and likely provide more robust insights.

Reviewer #2

(Remarks to the Author)

This manuscript makes some interesting observations regarding the mechanisms of sialic acid acetylation in mammals. This is an active area of research in the glycobiology field with important ramifications on our understanding of genetic disorders, cell signalling and host-pathogen interactions.

The main claims of this manuscript are:

- 1) SLC33A1 is a Golgi acetyl-CoA antiporter that provides most of the acetyl-CoA substrate to the luminal catalytic domain of CASD1 to acetylate CMP-sialic acid.
- 2) CASD1 has a catalytic transmembrane domain that enables SLC33A1-independent CMP-Sia acetylation using cytosolic pools of acetyl-CoA.

The first claim is well supported by the data presented. My main critique of this part of the manuscript is that it leans a little too heavily on AlphaFold2 models to interpret the consequences of disease-causing mutations on protein function (L227-274, Fig 3d, S8), especially given that the protein has been modeled as a monomer when it is claimed to exist as a homodimer within the membrane. Have the authors considered examining these mutations in the context of the dimer AF2/3 model? Regardless, this mutation analysis is conjectural and should be substantially trimmed down.

I am less convinced by the conclusions of the second part of the manuscript, which arguably makes the more interesting claim. The main findings in this section hinge upon data generated using two 'virolectins': BCoV-He0-Fc and ICVHe0-Fc. The latter is claimed to be specific for 9-O-Ac-Sia, while the former is claimed to recognize 7,9-di-O-Ac-Sia and, to a lesser extent, 9-O-Ac-Sia. The imperfect specificity of BCoV-He0-Fc makes interpreting this data difficult. It is unclear if weak signals observed with this lectin are due to weak recognition of 9-O-Ac-Sia, strong binding to a small amount of 7,9-di-O-Ac-Sia, or binding to some other uncharacterized glycoform (e.g. 7-O-Ac-Sia). An orthogonal means of detecting and quantifying

the glycoforms in these cells that is independent of the virolectins is required to provide greater confidence in the authors' interpretations of this data. MS-based glycomics seems well-suited to this task.

Further to this, stronger evidence is required before concluding that the CASD1 transmembrane domain has acetyltransferase activity. The claim is currently supported by partial structural homology to HGSNAT and OafB (as determined using AlphaFold2 models and without absolute conservation of active site residues), an MD simulation, and some virolectin data (which has its limitations, as discussed above). Direct detection and structural assignment of the products generated by CASD1 and its mutants (e.g. WT vs S49A vs H446A) *in vitro* would be far more convincing.

Regarding the MD simulation, it is not clear to me if the Sia O-9 is in close proximity to the acetyl group of acetyl-CoA, or not. If it is, this should be illustrated more clearly. At present I don't think it is. Indeed, when looking at Figure 6f, I wondered if this site might actually be catalyzing acetyl migration from O-9 to O-7, rather than performing an acetylation with Ac-CoA. Have the authors considered the possibility that the TMD of CASD1 is actually a 'migratase' rather than a transferase?

I think this is an interesting and important study and my hope is that the authors might be given the chance to further support their conclusions through the acquisition of more data using orthogonal techniques.

Some further comments/questions regarding figures:

Figure S5. Please quantitate and plot the degree of colocalization of these constructs over a larger number of cells. It appears to be imperfect, even for differently tagged versions of SLC33A1. Also, is B4GALT1 really in the same compartments as SLC33A1? It doesn't look like it is to me... and if not, is it really a useful control for PLA experiment?

Figure 3. What is the extra unassigned band in the western blot of Fig 3b?

Figure 7. In my view, there is no strong evidence that SLC33A1 forms a complex with CASD1. Co-localisation and PLA certainly aren't sufficient to make that claim. As such, it'd be best if they weren't represented this way in the concluding cartoon.

Figure 6d. WT control and H446A mutant needed.

Reviewer #3

(Remarks to the Author)

The manuscript by Malena Albers et al., "Interplay of SLC33A1-dependent and -independent Golgi sialic acid O-acetylation 1 in CASD1 catalysis", describes a novel biosynthetic pathway for generation of mono and di-O-acetylated sialic acids and sialoglycoconjugates containing these molecules. The authors demonstrate that the putative AcCoA transporter SLC33A1 is essential for the synthesis of GD3 ganglioside containing a N-acetyl-neuraminic acid residues bearing 9-O-acetylation catalyzed by the sialyltransferase ST8SIA1. The authors also demonstrate that the mutant SLC33A1 variants identified in patients with two rare neurological genetic disorders are inactive or only partially active. They further show that the acetylation reaction in the Golgi lumen is catalyzed by a luminal (LCD) domain of a multi-spanning transmembrane protein CASD1 with previously unknown function. The authors further show that this reaction requires a catalytic triad with a nucleophile Ser94, however, the Ser94Ala mutation that abolishes catalytic activity at the LCD site, does not block synthesis of mono- 9-O-acetylated sialosides. The authors explain this paradox by identifying the second active site (CTMR) situated in a transmembrane channel formed by the TM domains 5,6,7 and 13 of CASD1. Using structural modeling and dynamic simulations, they further demonstrate that the CTMR catalytic channel has a structural and a functional similarity with the catalytic channel of lysosomal N-acetyltransferase HGSNAT that catalyses transmembrane acetylation of lysosomal heparan sulfate using the cytoplasmic pool of AcCoA. Similar function and a structure has a catalytic channel of a bacterial O-acetyltransferase OafB. This reaction involves a nucleophile His residue essential for synthesis of di-7,9-O-acetylated sialosides but not of mono- 9-O-acetylated sialosides.

Overall, the results show that transmembrane acetylation is a common physiological process in multiple species including humans and that a synthesis of O-sialylated sialosides is a complex highly regulated process. The experiments are well designed and support main conclusion of the authors. The manuscript describes a novel biological phenomenon and is of potential interest for the broad readership of the journal, however, there are several points that need to be addressed.

First, most (if not all) experiments were conducted with immortalised cell lines overexpressing proteins of interests, thus it remains to be shown if the identified pathways are conserved in specific cells and tissues, especially in the neurons, where the gangliosides play essential roles. I suggest that the authors acknowledge this limitation in the end of the Discussion.

Second, the catalytic activities of LCD and CTMR domains need to be directly demonstrated *in vitro* using, for example, homogenates of CHO delta-Casd1 cells overexpressing WT and Ser/His CASD1 mutants, complemented with AcCoA and different sialosides as acceptors. Such assay is easily conducted for HGSNAT and has been essential to prove its enzymatic activity. Besides, such experiment would help to define the structural selectivity of the two active sites and whether the acetylation of the first site (presumably at 9-O position) by CTMR affects the acetylation at the O-7 position by LCD.

Finally, the authors postulate in the Discussion and show on the scheme in Fig.7 that SLC33A1 facilitates the transport of

CoA back to the cytoplasm. My impression is that this function has not been proven in the current work and remains to be a hypothesis. If so, this needs to be emphasized in the Discussion.

I also have some specific suggestions:

Figure 2h, PLA experiment. The images are small and do not allow to conclude in which cellular compartment the signal was detected. The signal is located in the perinuclear puncta which could be either Golgi or catabolic compartments such as lysosomes or autophagosomes. The cells need to be co-labeled for Golgi either using the same slides or separate slides co-labeled with a Golgi marker anti-Flag anti-V5 antibodies like in the figure S5

Figure 3b. What is the identity of a ~30 kDa band on the panel? Is it related (fragment) to SLC33A1 or just an unspecifically reacting protein? This needs to be explained in the figure legend.

Figure 4a. As in the Figure 2, Golgi localisation of proteins carrying mono and di-O-acetylated sialic acids needs to be confirmed by co-labeling with a Golgi marker. Moreover it would be interesting to compare images of permeabilized and non-permeabilized cells in panels b-d to show whether the proteins carrying 7,9-O-acetyl sialic acid are also present intracellularly.

The Figure 7 could be edited to include the synthesis of O-Ac-GD3 ganglioside by ST8Sia1.

Some readers would benefit from the explanation that viral lectins cannot recognize and bind mono and di-acetylated sialic acids in gangliosides and are, thus, specific for sialylated glycoproteins.

Version 1:

Reviewer comments:

Reviewer #1

(Remarks to the Author)

My concerns have been addressed in this revision and I do not have any further comments.

Reviewer #2

(Remarks to the Author)

I would like to thank the authors for their significant efforts in addressing the concerns I raised in my initial review. The revised manuscript is greatly improved and I think it makes a significant contribution to the glycobiology field.

I have a couple of comments:

1) It was unclear to me why disruption of Slc33a1 or the CASD1 LCD abrogates 9-O-Ac GD3 biosynthesis (as detected by TLC with an antibody to 9-O-Ac GD3) but does not disrupt 9-O-NeuAc (as detected by immunofluorescent microscopy with the virolectin ICV-HE0-185). Does this apparent discrepancy result from the different detection reagents involved (mAb vs virolectin)? Is there a reason why the TLCs are never developed using the virolectins? Using the same detection reagent in both assays could address this apparent contradiction.

2) The CoA detected with the in vitro assay in Fig 6f could be a result of hydrolysis rather than acetyl transfer. I accept that it is difficult to quantitate the acetylated CMP-NeuNAc but it would be helpful (and feasible) to show by MS that this species is forming in an enzyme-dependent manner within the samples. This in vitro demonstration of enzyme activity would strengthen the most significant claim of the manuscript (i.e. that CASD1 is a bifunctional enzyme).

Reviewer #3

(Remarks to the Author)

The authors have been very responsive in addressing the reviewers comments and suggestions and the revised manuscript has been greatly improved compared to the original draft. This reviewer also appreciates the difficulty of analyzing O-acetylated O-glycans in complex biological systems due to their extreme lability, which makes some experiments suggested by the reviewers unfeasible with the currently available Glycomics methodology. However, the recently released cryo-EM structure of SLC33A1 provides now additional support for the main conclusions of the manuscript. I do not have additional comments or concerns.

Version 2:

Reviewer comments:

Reviewer #2

(Remarks to the Author)

The authors are to be commended for repeating the experiment in Fig. 5d with both UM4D4 and ICV-HE-Fc to ensure methodological consistency. This demonstrates key differences in what the detection reagents recognize. Given how important these detection reagents are in formulating the conclusions of the paper, I don't understand why the authors would want to keep it in the rebuttal only. Please add it to the SI or manuscript so the reader can appreciate the differences between these critical reagents.

Regarding the in vitro assay, the authors state: "We believe that a definitive in vitro demonstration would require substantial optimization—including purification of highly active recombinant CASD1, refinement of reaction conditions (e.g., pH buffering, stabilizing agents), and the use of advanced analytical platforms." I don't disagree with this statement but isn't the point of this work to provide a definitive demonstration of the enzyme activity rather than just suggestive data?

REVIEWER COMMENTS

We would like to sincerely thank the reviewers for their careful reading of our manuscript and for providing thoughtful and constructive comments. We greatly appreciate the time and effort invested in reviewing our work, which has helped us improve both the clarity and quality of the manuscript. We have carefully considered each comment and made revisions to address the concerns raised. In the following, we provide detailed, point-by-point responses to all comments, highlighting the changes made in the revised manuscript where appropriate.

Point-by-point response:

Reviewer #1 (Remarks to the Author):

This manuscript by Albers et al. focuses on a membrane protein sialate O-acetyltransferase CAS1 domain-containing 1 (CASD1). The primary finding—that CASD1 possesses two active sites, one dependent on SLC33A1 (acetyl-CoA transporter) and one independent—is exciting and significantly advances our understanding of this enzyme and Golgi acetylation processes. The manuscript is well-structured, with a clear presentation of results, and this reviewer identifies no major concerns. I strongly support its publication in Nature Communications, with some minor concerns detailed below.

We are grateful to Reviewer 1 for the encouraging comments.

1. In Figure 2f, only the Ex 1-5 del clone is presented. Why were the other two Ex clones omitted? Including all three would allow readers to better assess clonal variation in the data. Additionally, could the authors clarify when acetyl-CoA levels were measured and whether these measurements align with the time points in Figure 2e?

The acetyl-CoA levels were measured 24 hours after cell seeding, which corresponds to the first time point presented in Fig. 2e. Information on the timing is described in the Methods section and is now also provided in the legend of Fig. 2 for clarity.

The choice of the Ex 1-5 del clone for the acetyl-CoA measurement was guided by the chronological sequence of our study and the rationale underlying the generation of this particular clone. Upon identifying residual Sia O-acetylation in our initial Δ *Slc33a1* clones, Ex1 and Ex3, we wondered whether exon skipping or the presence of an unannotated alternative upstream exon enabled the formation of functional translation products. To rigorously exclude this possibility, we generated the Ex 1-5 del clone, characterized by a large deletion encompassing most of the *Slc33a1* locus. Given this, the majority of our subsequent experiments were performed with the Ex 1-5 del clone. The acetyl-CoA measurements were conducted at a later stage of our study and at that timepoint, we determined that analysing the Ex 1-5 del clone would be sufficient to exclude acetyl-CoA shortage as a potential cause of the observed loss-of-function phenotype.

2. In Figure 3b, the anti-Flag western blot shows an impurity band near the Flag-SLC33A1-Y336* truncated protein. Could the authors clarify its identity or source? The band's intensity appears correlated with ST8SIA1 expression.

We sincerely thank all three Reviewers for identifying the omitted labelling of nonspecific bands in the anti-Flag Western blot shown in Fig. 3b. We observed, indeed, two non-specific bands at approximately 30 kDa and 50 kDa, which appeared consistently across all lanes, including the empty vector control.

In the revised version of Fig. 3b, these bands have been clearly marked with asterisks and are designated as nonspecific in the updated figure legend. Moreover, we performed an additional control experiment, in which we switched the used epitope tags and co-expressed V5-tagged SLC33A1 variants with Flag-tagged ST8SIA1. The corresponding anti-V5 Western blot, which lacks nonspecific bands, is presented in the new Supplementary Figure 6.

In the Results section, we added the following paragraph:

“The anti-Flag antibody used showed two non-specific bands (marked by asterisks in Fig. 3b), with the lower band migrating close to the truncated variant p.Y366*. For clearer visualization, we analyzed additionally V5-tagged constructs, as shown in Supplementary Fig. 6.”

3. The MD simulation details should be included in the main methods section rather than relegated to the supplementary material. Given the complexity of the simulation system, the energy minimization and equilibration steps seem brief. Furthermore, the production phase of only 100 ns is relatively short; with current computational resources, extending it to 1 μ s would be feasible and likely provide more robust insights.

We thank Reviewer 1 for this valuable suggestion. We have now completed 2000 ns of MD calculations in triplicate. Overall, we conclude the same outcome. The new data set is shown in the revised version of Supplementary Fig. 17. A converged structure snapshot after 2000 ns MD simulation is shown in the revised version of Fig. 6d. The description of the MD simulation details have been transferred to the main methods section as suggested. The paragraph describing the MD simulation in the Results section has been revised and now reads as follows.

“Closer examination of the CASD1 model revealed that H446 is situated at the bottom of a luminal pocket that could serve as acceptor binding site. A structure-guided approach was then used to derive an initial CTMR–CMP–Neu5Ac complex, in which we sought to test the stability and orientation preference of the acceptor substrate, using molecular dynamics (MD) simulations. The initial complex, embedded in a 1,2-dipalmitoyl-*sn*-glycero-3-phosphocholine (DPPC) lipid bilayer model, was used to perform three independent MD simulations that commenced from a different configuration and different initial velocity distributions (see Supplementary Figure 17 for details). The converged structure snapshot after 2000 ns MD simulation is shown in Fig. 6d. From this simulation, we observed that CMP–Neu5Ac occupies the luminal binding pocket in an orientation that positions the 9'-hydroxyl group of CMP–Neu5Ac within hydrogen bonding distance to H446 (see further detail in Supplementary Figure 17d). This simple MD model supports the notion that the CTMR of CASD1 can accommodate CMP–Neu5Ac in an orientation that allows *O*-acetylation via H446, possibly by activating the 9'-hydroxyl group for a nucleophilic attack on the carbonyl group of acetyl-CoA.”

Reviewer #2 (Remarks to the Author):

This manuscript makes some interesting observations regarding the mechanisms of sialic acid acetylation in mammals. This is an active area of research in the glycobiology field with important ramifications on our understanding of genetic disorders, cell signalling and host-pathogen interactions.

The main claims of this manuscript are:

- 1) SLC33A1 is a Golgi acetyl-CoA antiporter that provides most of the acetyl-CoA substrate to the luminal catalytic domain of CASD1 to acetylate CMP-sialic acid.
- 2) CASD1 has a catalytic transmembrane domain that enables SLC33A1-independent CMP-Sia acetylation using cytosolic pools of acetyl-CoA.

The first claim is well supported by the data presented. My main critique of this part of the manuscript is that it leans a little too heavily on AlphaFold2 models to interpret the consequences of disease-causing mutations on protein function (L227-274, Fig 3d, S8), especially given that the protein has been modeled as a monomer when it is claimed to exist as a homodimer within the membrane. Have the authors considered examining these mutations in the context of the dimer AF2/3 model? Regardless, this mutation analysis is conjectural and should be substantially trimmed down.

During the revision of our manuscript, a first cryo-EM structure of human SLC33A1 in complex with acetyl-CoA has been solved (PDB ID 9M0S). The corresponding paper includes an analysis of the SPG42 mutation S113R, while other patient-derived mutations are not mentioned (Zhou et al. 2025 Cell Discov. 11:36; doi: 10.1038/s41421-025-00793-1).

Instead of relying on AlphaFold models, we now map the mutations A110P, G509S and Y366* onto the cryo-EM structure of SLC33A1 and discuss their potential consequences on the basis of the experimental structure. With regard to the S113R mutation, we refer to Zhou et al. 2025. The original paragraph in the Results section has been replaced with a newly written version (see page 11 of the marked-up version). In the revised Fig. 3, we now show a cartoon representation of the cryo-EM structure of SLC33A1 instead of an AF2 model. The former Supplementary Fig. 8 (AF2 models of SLC33A1 variants) has been deleted. Furthermore, we have removed the former Supplementary Fig. 7, which provided evidence for the cytosolic orientation of the N- and C-termini of SLC33A1 through selective membrane permeabilization, as this information is now comprehensively established by the recently published cryo-EM structure of SLC33A1.

With regard to the oligomerization state of SLC33A1, the following points can be stated. Using analytical ultracentrifugation of solubilized SLC33A1, Peng and co-workers observed SLC33A1 homodimers (Peng et al. 2014 J. Neurosci. 34:6772; doi: 10.1523/JNEUROSCI.0077-14.2014), while the cryo-EM map of SLC33A1, like its AF2 model, shows a monomer. A similar discrepancy has been observed for the human dopamine transporter SLC6A3 (PDB ID 9EO4; Das et al. 2019 JBC 294:P5632; doi: 10.1074/jbc.RA118.006178). As shown by Chadda et al., transporter dimerization can be driven solely by differences in lipid solvation energetics between monomeric and dimeric states, without requiring specific protein-protein interactions (Chadda et al 2021 ELife 10:e63288; doi: 10.7554/eLife.63288). In such cases, the solubilization environment might become critical in determining whether the transporter is observed as a monomer or a dimer.

I am less convinced by the conclusions of the second part of the manuscript, which arguably makes the more interesting claim. The main findings in this section hinge upon data generated using two 'virolectins': BCoV-He0-Fc and ICVHe0-Fc. The latter is claimed to be specific for 9-O-Ac-Sia, while the former is claimed to recognize 7,9-di-O-Ac-Sia and, to a lesser extent, 9-O-Ac-Sia. The imperfect specificity of BCoV-He0-Fc makes interpreting this data difficult. It is unclear if weak signals observed with this lectin are due to weak recognition of 9-O-Ac-Sia, strong binding to a small amount of 7,9-di-O-Ac-Sia, or binding to some other uncharacterized glycoform (e.g. 7-O-Ac-Sia). An orthogonal means of detecting and quantifying the glycoforms in these cells that is independent of the virolectins is

required to provide greater confidence in the authors' interpretations of this data. MS-based glycomics seems well-suited to this task.

We sincerely appreciate the reviewer's insightful comments and constructive feedback. Virolectins are currently the most powerful tools for studying *O*-acetylated sialosides due to their exceptional selectivity and the critical advantage that they do not require sample processing steps that risk de-*O*-acetylation. The binding specificities of these lectins have been extensively characterized by the groups of Raoul de Groot and Geert-Jan Boons, who developed a comprehensive library of synthetic *O*-acetylated sialosides covering nearly all possible acetylation patterns (Li et al., Nat. Chem. 2021, 13:496; doi:10.1038/s41557-021-00655-9). Their work definitively established that 9-*O*-acetylation is an absolute requirement for binding by BCoV-HE⁰-Fc and ICV-HE⁰-Fc, while 7-mono-*O*-acetylated sialosides are not recognized. Furthermore, they formally demonstrated a strong preference of BCoV-HE⁰-Fc for 7,9-di-*O*-acetylated sialosides over their 9-mono-*O*-acetylated counterparts. We fully agree that the development of a lectin specifically selective for 7,9-di-*O*-acetylated sialosides would represent a significant advance in the field and greatly enhance our ability to study this biologically relevant glycoform. However, the absence of a reagent with exclusive specificity for 7,9-di-*O*-acetylation remains a notable gap in the field.

We also thank the reviewer for the suggestion to employ MS-based glycomics to further characterize the *O*-acetylated glycoforms in our system. While we would have been eager to pursue this approach, the proposed analysis presents substantial technical challenges that extend well beyond the scope of the current study. Specifically, our work focuses on the enzymatic formation of 9- and 7,9-*O*-acetylated sialosides in the context of an *O*-glycan-specific sialyltransferase. This would necessitate the analysis of *O*-acetylated *O*-linked glycans attached to as-yet-unidentified carrier proteins—a task that demands specialized glycomics workflows not routinely available in standard laboratories.

After discussing with experts in the field of MS-based glycan analytics, it became clear that such an analysis would require (i) non-standard glycomics, (ii) extensive method development to ensure sensitivity and specificity; and (iii) timely data interpretation by a specialist in glycomics. Moreover, even with these efforts, the inherent instability of *O*-acetyl groups poses a fundamental obstacle. These groups are highly susceptible to spontaneous loss, migration, and hydrolysis during sample handling, chromatography, and MS/MS fragmentation (Refs. 6, 56). As a result, *O*-acetylated species are frequently underrepresented or misassigned, and positional isomers—such as 7-*O*-acetyl versus 9-*O*-acetyl—cannot be reliably distinguished using conventional fragmentation techniques.

This challenge is further exacerbated by the conditions typically used in glycomics workflows. For instance, *O*-glycan release via base-mediated β -elimination—commonly employed in glycomics—inevitably leads to the hydrolysis of acetyl esters. Even when employing advanced fragmentation methods such as ETHcD for glycopeptide analysis, only a small number of *O*-acetylated *O*-glycan structures have been confidently assigned to date (see Ref. 42, 43; Pap et al. 2020 Mol. Omics, 16:156). Notably, these structures have been identified primarily on proteins for which *O*-glycosylated peptides can be obtained that are particularly amenable to MS analysis—namely, those with high ionization efficiency and only a single *O*-glycosylation site—representing an additional constraint.

While new analytical methods are emerging, they have been successfully applied primarily to well-characterized, highly *O*-acetylated model proteins—such as bovine submandibular mucin (Vos et al., 2023, Nat. Commun. 14:6795). This underscores the broader difficulty of analyzing *O*-acetylated *O*-glycans in complex biological systems, where these glycoforms are less abundant. Taken together, MS-

based detection and structural characterization of *O*-acetylated sialic acids—both in general and specifically on *O*-glycans—remain significant and unresolved analytical challenges.

Further to this, stronger evidence is required before concluding that the CASD1 transmembrane domain has acetyltransferase activity. The claim is currently supported by partial structural homology to HGSNAT and OafB (as determined using AlphaFold2 models and without absolute conservation of active site residues), an MD simulation, and some virolectin data (which has its limitations, as discussed above). Direct detection and structural assignment of the products generated by CASD1 and its mutants (e.g. WT vs S49A vs H446A) in vitro would be far more convincing.

We thank Reviewer 2 for this insightful comment. In response, we have now provided (i) direct biochemical evidence for the acetyltransferase activity of the C-terminal transmembrane region (CTMR) of CASD1 through an in vitro assay, and (ii) additional functional validation of the catalytic transmembrane tunnel via an extensive mutational analysis of newly generated single and double mutants.

(i) To directly assess acetyltransferase activity, we adapted a thiol-based assay to quantify the release of free CoA (HS-CoA) from acetyl-CoA during the transfer of an acetyl group to CMP-Neu5Ac. This approach was inspired by a recently published method for HGSNAT (Zhao et al., 2024, Nat Commun 15:5388) and proved robust and reliable under our experimental conditions. In contrast, an alternative HPLC-based method intended to detect the *O*-acetylated product, CMP-Neu5,9Ac₂, failed to yield sensitive results. Although chromatographic separation of the standards (CMP-Neu5Ac and CMP-Neu5,9Ac₂) was successful, the method lacked sufficient sensitivity to detect the low micromolar concentrations of product generated—levels that were confidently quantified using the thiol assay. Prolonging reaction times to enhance product accumulation was not feasible, as CMP-Neu5Ac is chemically labile under the assay conditions (pH 6.5, 37°C), undergoing self-degradation via a phosphate–carboxylate anhydride intermediate (Kajihara et al., Chem. Eur. J. 2011, 17:7645). Under the conditions used, a substantial amount of the input CMP-Neu5Ac (500 μM) hydrolyzed to CMP and Neu5Ac within 3 hours.

(ii) Guided by structural superposition of our CASD1 model with the cryo-EM structure of HGSNAT, we targeted conserved residues lining the predicted transmembrane catalytic tunnel. We generated a panel of single and double alanine mutants, including combinations with the S94A mutation to inactivate the luminal catalytic domain (LCD), thereby isolating the contribution of the CTMR. As shown in the new Figure 6g, five residues within the CTMR were found to be essential for 9-*O*-acetylation of sialic acid—phenotypes that were only fully apparent in the absence of functional LCD activity. These findings provide strong functional support for the structural and mechanistic parallels between CASD1 and HGSNAT, and collectively reinforce the conclusion that the CTMR of CASD1 functions as a transmembrane acetyltransferase.

Together, these new data offer compelling biochemical and genetic evidence for the catalytic role of the CASD1 CTMR and further validate the proposed architecture of its transmembrane tunnel. The data are presented in the new Figs. 6e-i and Supplementary Figs. 18-21, and are described in the Result section (see pages 20-22 of the marked-up manuscript) as follows:

“To directly demonstrate that the CTMR of CASD1 functions as an *O*-acetyltransferase, we performed an on-bead enzyme assay using detergent-solubilized, Myc-V5-tagged CASD1 immobilized on Myc nanobody-coated beads as enzyme source (Fig. 6e). After incubation with CMP-Neu5Ac and acetyl-

CoA, we quantified the acetyltransferase reaction product HS-CoA using a thiol-sensitive dye, as described recently for HGSNAT⁴⁸. Under the used conditions, CASD1-S94A retained over 40% of the WT activity (Fig. 6f). Since this mutant lacks a functional LCD, the remaining SOAT activity can be attributed to the CTMR. Although only the double mutant S94A;H446A—harbouring defects in both LCD and CTMR—was expected to be inactive, the single mutant CASD1-H446A likewise showed no detectable activity. Effective enzyme loading was verified by Western blot analysis, as shown exemplarily for one experiment in Fig. 6f and for all biological replicates in Supplementary Fig. 18.

To circumvent potential destabilizing effects of detergent solubilization, we employed cellular complementation using CHO- Δ Casd1 cells to further assess the functional impact of CASD1 mutations. The ability of CASD1 variants to restore 9-*O*-acetylated sialosides upon co-expression with ST8SIA6 was analysed by flow cytometry using the ICV-derived virolectin (Fig. 6g; with dot plots and expression controls shown in Supplementary Figs. 19 and 20, respectively). In the cellular setting, CASD1-S94A retained >60% of the WT complementation activity. To definitively exclude any contribution from the mutated LCD, we replaced the entire domain with a FLAG epitope. The resulting deletion mutant, CASD1- Δ LCD, exhibited complementation activity comparable to CASD1-S94A, indicating that the CTMR alone accounts for the residual activity. CASD1-H446A, like the double mutant CASD1-S94A;H446A, reached no significant activity over the mock control. Likewise, substitution of H446 with alternative residues (H446Y/F/Q/S) invariably abolished complementation activity (Supplementary Fig. 21). In HGSNAT, alanine replacement of N286, positionally equivalent to CASD1-H446 by structural superposition (see Fig. 6h), abolished enzymatic activity and caused abnormal oligomerization⁴⁸, highlighting a critical role of this position in protein folding and stability. Moreover, HGSNAT-N286 plays a pivotal functional role by stabilizing the acetyl group of acetyl-CoA within the active site^{47, 48, 49}. Based on the structural overlay, we identified CASD1-Q714 as putative functional equivalent of HGSNAT-N286, even though the two residues lie in distinct yet adjacent TMHs of the 9-TMH core (Fig. 6h). Q714 is strictly conserved across phylogenetic distant CASD1 orthologues (Supplementary Figure 16), and its replacement by alanine reduced complementation activity by ~35% (Fig. 6g). This loss increased to >90% when Q714A was combined with the LCD-inactivating S94A mutation (Fig. 6g; CASD1-S94A;Q714A), indicating a functional role of Q714 for CTMR-mediated *O*-acetylation.

Furthermore, as shown in Fig. 6i, we found that the four CASD1 residues K435, R428, R489 and R496 align structurally with the four positively charged HGSNAT residues that bind the adenosine diphosphate moiety of acetyl-CoA^{47, 48, 49}. In HGSNAT, these sites are essential (R275 and R267) or contribute significantly (R345 and K341) to enzymatic activity⁴⁸. Consistent with this, alanine substitution of the corresponding CASD1 residues severely impaired complementation activity when combined with the LCD-inactivating S94A mutation (Fig. 6g). In the presence of a functional LCD, however, the loss-of-function effect of the K435A, R428A, R489A and R496A mutations was largely masked by the activity of the LCD. Notably, the four positively charged residues identified as critical for CASD1's CTMR function are not only structurally congruent with the basic acetyl-CoA-binding residues of HGSNAT (Fig. 6i), but also with a quartet of basic residues in the OafB model (Supplementary Figure 22), further reinforcing the structural homology of the 9-TMH cores across HGSNAT, CASD1 and OafB."

Regarding the MD simulation, it is not clear to me if the Sia O-9 is in close proximity to the acetyl group of acetyl-CoA, or not. If it is, this should be illustrated more clearly. At present I don't think it is.

We sincerely thank Reviewer 2 for this comment. Based on the suggestion of Reviewer 1, we have repeated the MD simulation by extending the production phase to 2000 ns. In Fig. 6d, we now present a converged structure snapshot of the 2000 ns MD simulation. For clarity, we present an enlarged image of the snapshot in the Supplementary Fig. 17d, illustrating H-bonding distance (2.2 Å) between Nδ1 of His446 and the acceptor hydroxyl group in position C9 of the Neu5Ac moiety.

Indeed, when looking at Figure 6f, I wondered if this site might actually be catalyzing acetyl migration from O-9 to O-7, rather than performing an acetylation with Ac-CoA. Have the authors considered the possibility that the TMD of CASD1 is actually a 'migratase' rather than a transferase?

We thank Reviewer 2 for this interesting thought. We cannot formally exclude the possibility that the transmembrane catalytic site facilitates acetyl migration from O-9 to O-7, a step that occurs at low frequency already non-enzymatically (Ref. 56). The structural similarity with HGSNAT, however, suggests that a role as transmembrane acetyltransferase remains a plausible function, in particular in light of the new data presented in Fig. 6g, as outlined in more detail above.

I think this is an interesting and important study and my hope is that the authors might be given the chance to further support their conclusions through the acquisition of more data using orthogonal techniques.

We sincerely appreciate Reviewer 2's encouraging feedback and hope that our new data satisfactorily resolve the reviewer's concern.

Some further comments/questions regarding figures:

Figure S5. Please quantitate and plot the degree of colocalization of these constructs over a larger number of cells. It appears to be imperfect, even for differently tagged versions of SLC33A1. Also, is B4GALT1 really in the same compartments as SLC33A1? It doesn't look like it is to me... and if not, is it really a useful control for PLA experiment?

We acknowledge that under transient overexpression conditions, certain constructs exhibited incomplete Golgi localization, accompanied by significant ER retention. To address this issue, we aimed at establishing stable co-transfectants. Despite extensive optimization of transfection conditions, antibiotic selection protocols, and clonal expansion, the co-expression of the plasmids—particularly those encoding the multi-pass transmembrane proteins SLC33A1 and CASD1—proved highly inefficient. This was reflected in the eventual undetectable expression levels of the multi-pass constructs in stable lines. In contrast, the single-pass transmembrane construct B4GALT1 was consistently amenable to stable integration and selection, yielding high-level, uniform expression in 80–100% of cells, with exclusive localization to the Golgi apparatus. These findings highlight the inherent difficulties in maintaining stable co-expression of multi-pass transmembrane proteins, due to their hydrophobic nature, complex folding requirements and low natural abundance.

In response to the reviewer's comment, we acknowledge that we were unable to fully resolve the concerns regarding the PLA experiment. As a result, we have removed the PLA data from the manuscript. To address the question of subcellular localization, we have now included new data demonstrating the colocalization of transiently expressed SLC33A1 and CASD1 with a validated Golgi marker (Fig. 2h), which provides clear and consistent evidence of their shared Golgi localization. We have revised the Result section and the Discussion section accordingly (see below).

Result section (page 10 of the marked-up manuscript):

“To examine whether the two proteins also target to the same compartment, we investigated their subcellular localization by indirect immunofluorescence analysis upon expression of V5-tagged constructs in CHO cells. As shown previously²⁰, CASD1 colocalizes well with the Golgi marker α -mannosidase II (Fig. 2h, upper panel). Although SLC33A1 displays a more diffuse perinuclear distribution, it nonetheless shows substantial colocalization with the marker protein (Fig. 2h, lower panel), indicating that a significant fraction of the SLC33A1 pool resides in the Golgi apparatus. Our finding contrasts with the strict ER localization observed by subcellular fractionation of CHO cells²⁵, but is consistent with immunofluorescence microscopy data demonstrating colocalization of SLC33A1 with the Golgi marker proteins giantin and syntaxin 6 in MCF-7 cells³⁰.”

Discussion section (page 23 of the marked-up manuscript)

“In contrast to the previous assumption that acetyl-CoA reaches the Golgi via vesicular transport from the ER⁵⁴, the Golgi localization of SLC33A1 and CASD1 observed in the present study, consistent with findings by Huppke et al.³⁰, suggest that acetyl-CoA is translocated directly from the cytosol to the Golgi lumen. The positioning of the acetyl-CoA transporter and the *O*-acetyltransferase within the same compartment ensures the effective utilization of imported acetyl-CoA by the LCD of CASD1.”

Figure 3. What is the extra unassigned band in the western blot of Fig 3b?

We appreciate the reviewer’s careful inspection of Fig. 3b. This point has been addressed in the revised version of our manuscript as outlined in our response to Reviewer 1 (see point 2).

Figure 7. In my view, there is no strong evidence that SLC33A1 forms a complex with CASD1. Co-localisation and PLA certainly aren’t sufficient to make that claim. As such, it’d be best if they weren’t represented this was in the concluding cartoon.

We appreciate this comment and have changed the concluding cartoon accordingly.

Figure 6d. WT control and H446A mutant needed.

As outlined above, we have expanded our analysis by generating a larger panel of CASD1 mutants. In the revised manuscript, the expanded panel is shown in Fig. 6g and includes WT control and H446A mutant.

Reviewer #3 (Remarks to the Author):

The manuscript by Malena Albers et al., “Interplay of SLC33A1-dependent and -independent Golgi sialic acid *O*-acetylation 1 in CASD1 catalysis”, describes a novel biosynthetic pathway for generation of mono and di-*O*-acetylated sialic acids and sialoglycoconjugates containing these molecules. The authors demonstrate that the putative AcCoA transporter SLC33A1 is essential for the synthesis of GD3 ganglioside containing a *N*-acetyl-neuraminic acid residues bearing 9-*O*-acetylation catalyzed by the sialyltransferase ST8SIA1. The authors also demonstrate that the mutant SLC33A1 variants identified in patients with two rare neurological genetic disorders are inactive or only partially active. They further show that the acetylation reaction in the Golgi lumen is catalyzed by a luminal (LCD) domain of a multi-spanning transmembrane protein CASD1 with previously unknown function. The authors further show that this reaction requires a catalytic triad with a nucleophile Ser94, however, the

Ser94Ala mutation that abolishes catalytic activity at the LCD site, does not block synthesis of mono-9-O-acetylated sialosides. The authors explain this paradox by identifying the second active site (CTMR) situated in a transmembrane channel formed by the TM domains 5,6,7 and 13 of CASD1. Using structural modeling and dynamic stimulations, they further demonstrate that the CTMR catalytic channel has a structural and a functional similarity with the catalytic channel of lysosomal N-acetyltransferase HGSNAT that catalyses transmembrane acetylation of lysosomal heparan sulfate using the cytoplasmic pool of AcCoA. Similar function and a structure has a catalytic channel of a bacterial O-acetyltransferase OafB. This reaction involves a nucleophile His residue essential for synthesis of di-7,9-O-acetylated sialosides but not of mono-9-O-acetylated sialosides. Overall, the results show that transmembrane acetylation is a common physiological process in multiple species including humans and that a synthesis of O-sialylated sialosides is a complex highly regulated process. The experiments are well designed and support main conclusion of the authors. The manuscript describes a novel biological phenomenon and is of potential interest for the broad readership of the journal, however, there are several points that need to be addressed.

First, most (if not all) experiments were conducted with immortalised cell lines overexpressing proteins of interests, thus it remains to be shown if the identified pathways are conserved in specific cells and tissues, especially in the neurons, where the gangliosides play essential roles. I suggest that the authors acknowledge this limitation in the end of the Discussion.

We thank Reviewer 3 for this suggestion and have now added the following statement to the Discussion (see page 25 of the marked-up manuscript):

“Future work will require validation in more physiologically relevant models, such as primary cells or animal systems, as the current study relies on cell lines and protein overexpression.”

Second, the catalytic activities of LCD and CTMR domains need to be directly demonstrated in vitro using, for example, homogenates of CHO delta-Casd1 cells overexpressing WT and Ser/His CASD1 mutants, complemented with AcCoA and different sialosides as acceptors. Such assay is easily conducted for HGSNAT and has been essential to prove its enzymatic activity. Besides, such experiment would help to define the structural selectivity of the two active sites and whether the acetylation of the first site (presumably at 9-O position) by CTMR affects the acetylation at the O-7 position by LCD.

The catalytic activity of the luminal catalytic domain (LCD) has been directly demonstrated in our previous study using an in vitro assay, employing the isolated LCD as the enzyme source (Baumann et al. 2015 Nat. Commun. 6:7673). As detailed in our response to Reviewer 2, we now provide direct experimental evidence for O-acetyltransferase activity associated with the C-terminal transmembrane region (CTMR) of CASD1. In accordance with Reviewer 3’s suggestion, we performed an on-bead enzyme assay using detergent-solubilized, Myc-V5-tagged CASD1 immobilized on Myc nanobody-coated beads as the enzyme source (see schematic in new Fig. 6e). Following incubation with CMP-Neu5Ac and acetyl-CoA, the reaction product, HS-CoA, was quantified using a thiol-sensitive dye, as recently established for HGSNAT (Ref. 49). Under these conditions, CASD1-S94A, which lacks a functional LCD, retained over 40% of the WT activity (Fig. 6f). This residual activity therefore unambiguously demonstrates that the CTMR possesses intrinsic O-acetyltransferase function.

However, determining the precise regioselectivity of acetylation by the two catalytic domains remains a significant analytical challenge. The O-acetyl group on sialic acid is inherently labile and prone to spontaneous migration along the glycerol side chain during sample preparation—particularly under

alkaline and acidic conditions or during derivatization steps. In our earlier work (Ref. 20), we established that the LCD transfers an acetyl group from acetyl-CoA to CMP-Neu5Ac, yielding CMP-Neu5,9Ac₂. This product was analyzed by DMB-HPLC, a method that involves acid-induced release of Sia from CMP-Sia and subsequent DMB-derivatization of the free reducing end—processes that are known to promote acetyl group loss and migration. Importantly, because acetyl-group migration can mask the original site at which the acetyl group is introduced, this phenomenon inherently limits our ability to determine the regioselectivity of the two catalytic domains of CASD1 with complete certainty. To address this limitation in our current study, we have tried to incorporate a more direct approach for the analysis of the CTMR of CASD1. [editorial note: confidential, unpublished information redacted] We employed an HPLC-based assay to monitor O-acetylated CMP-Sia directly. While this method enabled adequate separation of the standards CMP-Neu5Ac and CMP-Neu5,9Ac₂, it lacked the sensitivity required to detect the low micromolar levels of product generated under our assay conditions—levels that were readily quantifiable via the thiol-based assay for HS-CoA. Extending reaction times to increase product accumulation was not feasible, as CMP-Neu5Ac is chemically unstable under the reaction conditions (pH 6.5, 37°C), undergoing self-degradation via a phosphate–carboxylate anhydride intermediate (Kajihara et al., Chem. Eur. J. 2011, 17:7645). Under these conditions, a substantial proportion of the input CMP-Neu5Ac (500 μM) hydrolyzed to CMP and Neu5Ac within 3 hours.

Finally, the authors postulate in the Discussion and show on the scheme in Fig.7 that SLC33A1 facilitates the transport of CoA back to the cytoplasm. My impression is that this function has not been proven in the current work and remains to be a hypothesis. If so, this needs to be emphasized in the Discussion.

We thank Reviewer 3 for raising this point, which helped us identify and address missing information in the manuscript. SLC33A1's function as an antiporter has been demonstrated previously by the group of Luigi Puglielli (see Ref. 25 and 26). We have now added this information in the Introduction section (see page 5 of the marked-up version) as outlined below.

“In its solubilized state, the transporter exists as a dimer and exhibits functional characteristics consistent with an acetyl-CoA/CoA antiporter^{25, 26}.”

I also have some specific suggestions:

Figure 2h, PLA experiment. The images are small and do not allow to conclude in which cellular compartment the signal was detected. The signal is located in the perinuclear puncta which could be either Golgi or catabolic compartments such as lysosomes or autophagosomes. The cells need to be co-labeled for Golgi either using the same slides or separate slides co-labeled with a Golgi marker anti-Flag anti-V5 antibodies like in the figure S5

We thank Reviewer 3 for this comment. As explained in our response to Reviewer 2, we have removed the PLA data from the manuscript. To address the question of subcellular localization, we have now included new data demonstrating the colocalization of transiently expressed SLC33A1 and CASD1 with a validated Golgi marker (Fig. 2h), which provides clear and consistent evidence of their shared Golgi localization.

Figure 3b. What is the identity of a ~30 kDa band on the panel? Is it related (fragment) to SLC33A1 or just an unspecifically reacting protein? This needs to be explained in the figure legend.

We have addressed this point as explained in our response to point 2 of Reviewer 1.

Figure 4a. As in the Figure 2, Golgi localisation of proteins carrying mono and di-O-acetylated sialic acids needs to be confirmed by co-labeling with a Golgi marker.

We are grateful for this suggestion, which helps us to improve our manuscript. Golgi localization of mono- and di-O-acetylated sialoglycans is now demonstrated by co-localization with the Golgi marker α -mannosidase II. The new data set is presented in the new Supplementary Figure 9. In the Results section on page 14 of the marked-up manuscript, we added the following sentence:

“Golgi localization of the intracellular virolectin ligands was confirmed by co-staining with the Golgi marker α -mannosidase II (Supplementary Fig. 9).”

Moreover it would be interesting to compare images of permeabilized and non-permeabilized cells in panels b-d to show whether the proteins carrying 7,9-O-acetyl sialic acid are also present intracellularly.

For non-transfected CHO-WT cells, we have shown the presence of intracellular 7,9-O-acetylated sialosides in Fig. 4a (see upper left image). However, since the comment of Reviewer 3 may relate to comparing permeabilized and non-permeabilized cells after ST8SIA6 expression, we extended our intracellular analysis. The new data set is shown in the new Supplementary Figure 13 and includes virolectin stains of CHO cells of all three genotypes (WT, Δ *Slc33a1* and Δ *Casd1*) both before and after ST8SIA6 expression. The overall outcome corresponds to that of the cell surface staining shown in Figs. 4b-h, with the notable difference that intracellularly, CHO-WT cells—and to a certain extent also CHO- Δ *Slc33a1* cells—exhibit virolectin ligands already prior to ST8SIA6 expression. These ligands contribute to the overall ligand profile observed after ST8SIA6 expression, thereby complicating the interpretation of intracellular analyses relative to that of cell surface analyses. Identification of the sialyltransferase that drives the formation of the intracellular O-acetylated sialosides in non-transfected CHO-WT cells requires future work and was beyond the scope of the current study.

In the Results section (see page 16 of the marked-up version), we have added the following sentence: “Virolectin stains of permeabilized mock- and ST8SIA6-transfected cells are provided in Supplementary Fig. 13.”

Moreover, we have added the following explanatory note to the legend of Supplementary Fig. 13 for clarity:

“Please note that intracellular virolectin ligands present before ST8SIA6 expression (mock) contribute to the total staining signal detected following ST8SIA6 expression (+ ST8SIA6).”

The Figure 7 could be edited to include the synthesis of O-Ac-GD3 ganglioside by ST8Sia1.

We thank Reviewer 3 for this suggestion. However, including an additional scheme depicting the synthesis of O-Ac-GD3 in Fig. 7 would largely duplicate the information already presented in Fig. 2g. To

avoid redundancy while still facilitating comparison with Fig. 7, we have instead adapted Fig. 2g so that it now displays the luminal catalytic domain of CASD1 in the same format used for Fig. 7.

Some readers would benefit from the explanation that viral lectins cannot recognize and bind mono and di-acetylated sialic acids in gangliosides and are, thus, specific for sialylated glycoproteins.

As shown by Zimmer and co-workers, influenza C virus (ICV) binds to 9-*O*-acetylated sialic acids on both glycoproteins and gangliosides (Zimmer et al. 1992 Eur. J. Biochem. 204:209). Using the ICV-derived virolectin, we have obtained similar results in our own experiments (unpublished), further supporting the published data.

REVIEWER COMMENTS

We would like to sincerely thank the Editor and the Reviewers for their careful evaluation of our revised manuscript and for the constructive comments and suggestions. We greatly appreciate the time and effort invested in reviewing our work.

We are pleased that Reviewer 1 and 3 find that the revised version has satisfactorily addressed their comments. We are grateful for their positive feedback and thoughtful suggestions, which have helped us to improve the clarity and quality of the manuscript.

We sincerely thank Reviewer 2 for the additional comments. We have carefully considered these points and provide detailed, point-by-point responses below.

Point-by-point response:

Reviewer #1 (Remarks to the Author):

My concerns have been addressed in this revision and I do not have any further comments.

We thank the reviewer for confirming that the revised manuscript addresses all of their concerns.

Reviewer #2 (Remarks to the Author):

I would like to thank the authors for their significant efforts in addressing the concerns I raised in my initial review. The revised manuscript is greatly improved and I think it makes a significant contribution to the glycobiology field.

We thank the reviewer for their positive assessment of the revised manuscript and for recognizing its contribution to the field.

I have a couple of comments:

1) It was unclear to me why disruption of Slc33a1 or the CASD1 LCD abrogates 9-O-Ac GD3 biosynthesis (as detected by TLC with an antibody to 9-O-Ac GD3) but does not disrupt 9-O-NeuAc (as detected by immunofluorescent microscopy with the virolectin ICV-HE0-185). Does this apparent discrepancy result from the different detection reagents involved (mAb vs virolectin)? Is there a reason why the TLCs are never developed using the virolectins? Using the same detection reagent in both assays could address this apparent contradiction.

We sincerely thank the reviewer for this important comment, which allowed us to clarify this point. Notably, the two assays employ distinct sialyltransferase, ST8SIA1 and ST8SIA6, reflecting different glycan biosynthetic pathways. We address the concern in two parts: firstly, the detection reagents; and secondly, the functional implications of the two sialyltransferases.

1. Detection reagents

At the outset of the study, we focused on ST8SIA1-dependent synthesis of 9-*O*-acetylated GD3 (9-*O*-Ac-GD3), for which we used the well-characterized, commercially available monoclonal antibody UM4D4, which specifically recognizes 9-*O*-Ac-GD3. In contrast, virolectins such as ICV-HE⁰-Fc are not commercially available and were developed in-house during the course of the study. Notably, ICV-HE⁰-Fc, which recognizes 9-*O*-acetylated sialic acids in both α 2,3- and α 2,8-linkages, was not available at the initial stage and was introduced later to detect 9-*O*-acetylated sialoglycoproteins produced by the O-glycan-specific ST8SIA6.

In response to the reviewer's suggestion, we have repeated the experiment shown in Fig. 5d using both UM4D4 and ICV-HE⁰-Fc in parallel, ensuring methodological consistency. The results confirm that the luminal catalytic domain (LCD) of CASD1 is essential for ST8SIA1-dependent 9-*O*-Ac-GD3 synthesis, as evidenced by the loss of the 9-*O*-Ac-GD3 signal with the CASD1-S94A mutant. This is now shown by staining with UM4D4 and ICV-HE⁰-Fc (see Fig. A). Importantly, ICV-HE⁰-Fc detected not only 9-*O*-Ac-GD3 but also additional 9-*O*-acetylated ganglioside species (marked by asterisks), indicating broader reactivity. Given that the identity of these additional ganglioside species remains to be fully characterized, and considering that their biosynthesis may involve distinct sialyltransferases, we have chosen to present this supplementary data exclusively in the rebuttal letter. This allows us to address the reviewer's concern while preserving the focus of the manuscript.

Fig. A: Immuno-TLC of total gangliosides extracted from CHO- Δ Casd1 cells complemented with CASD1-WT or CASD1-S94A. CHO- Δ Casd1 cells were transiently transfected with a plasmid encoding ST8SIA1 in combination with a plasmid encoding either CASD1-WT or CASD1-S94A. Extracted gangliosides were separated by TLC and plates were stained with anti-GD3 antibody R24, anti-9-*O*-Ac-GD3 antibody UM4D4 or the virolectin ICV-HE⁰-Fc. Bands detected by ICV-HE⁰-Fc but not UM4D4 are marked by asterisks

2. Sialyltransferases

We propose that the apparent discrepancy in the observed product profiles stem from distinct substrate preferences of ST8SIA1 and ST8SIA6. ST8SIA6 generates 9-*O*-acetylated and 7,9-di-*O*-acetylated sialosides. It may thus preferentially utilize 9-*O*-acetylated and 7,9-di-*O*-acetylated CMP-sialic acid (CMP-Sia), produced by the CTMR of CASD1 and the concerted action of LCD and CTMR, respectively. According to this model, disruption of *Slc33a1* or the CASD1 LCD would abolish 7,9-di-*O*-acetylated sialoside formation but preserve ST8SIA6-mediated 9-*O*-acetylation—which is consistent with our experimental data.

In light of prior evidence for the cellular occurrence of 7-*O*-acetylated GD3 (7-*O*-Ac-GD3) and the previously proposed CASD1/ST8SIA1-mediated formation of 7-*O*-Ac-GD3 (Erdmann et al. 2006 *Glycoconj. J.* 23:23; Arming et al. 2011 *Glycobiology* 21:553), it is plausible that ST8SIA1, in contrast, utilizes 7-*O*-acetylated CMP-Sia—generated by the LCD of CASD1—as a substrate to synthesize 7-*O*-Ac-GD3. This product can undergo spontaneous ester migration to form 9-*O*-acetylated GD3, a non-enzymatic process that occurs efficiently even at physiological pH (Zhang et al. 2026 *Angew. Chem. Int. Ed.* 65:e17989). According to this model, loss of *Slc33a1* or the CASD1 LCD would disrupt the initial 7-*O*-acetylation step, thereby abrogating the synthesis of 7-*O*-Ac-GD3 and, consequently, its downstream conversion to 9-*O*-Ac-GD3—which is consistent with the experimentally observed loss of 9-*O*-Ac-GD3.

Collectively, this conceptual model—based on differential substrate utilization by ST8SIA1 and ST8SIA6, coupled with the proposed regioselective functions of CASD1’s catalytic domains—provides a coherent and mechanistically plausible explanation for the observed phenotypic differences.

For clarity, we have revised the corresponding paragraph in the Discussion Section (page 23 of the revised main manuscript) and included a schematic of the proposed pathway for the two-step generation of 9-*O*-acetylated GD3 as new Supplementary Figure 23 (see below).

The revised paragraph in the Discussion Section now reads as follows.

“Our MD simulations on the CTMR model show binding of CMP-Neu5Ac in an orientation that would allow 9-*O*-, but not 7-*O*-acetylation, suggesting that the CTMR is selective for the C9 hydroxyl group, while the LCD mediates 7-*O*-acetylation. At first glance, this appears puzzling, since our data also demonstrated that the LCD, together with SLC33A1 and ST8SIA1, mediates the formation of 9-*O*-acetylated gangliosides. However, in light of prior evidence for the cellular occurrence of 7-*O*-acetylated GD3 (7-*O*-Ac-GD3)^{56, 57}, it is plausible that ST8SIA1 uses 7-*O*-acetylated CMP-Sia—produced by the LCD of CASD1—as a donor substrate, thereby initially forming 7-*O*-Ac-GD3. This product can subsequently undergo non-enzymatic conversion to 9-*O*-Ac-GD3, as the 7-*O*-acetyl group is prone to

migrate to C9, yielding a thermodynamically more stable primary ester bond^{58,59} (see Supplementary Fig. 23 for a schematic of the proposed pathway). ”

Supplementary Figure 23.

Proposed Model for the generation of 9-O-acetylated GD3. Based on prior evidence for 7-O-acetylated GD3 (7-O-Ac-GD3)^{17,18}, we propose a model, in which ST8SIA1 preferentially utilizes 7-O-acetylated CMP-Sia—generated by the LCD of CASD1—as a substrate, producing 7-O-Ac-GD3. Since the 7-O-acetyl group is prone to spontaneous migration to C9 of Sia, 7-O-Ac-GD3 is converted to 9-O-acetylated GD3 (9-O-Ac-GD3) over time¹⁹. This non-enzymatic process may take place within the Golgi apparatus or after transport to the plasma membrane (PM). According to this model, loss of SLC33A1 or inactivation of the CASD1 LCD through the S94A mutation disrupts the initial 7-O-acetylation step, thereby preventing the downstream production of 9-O-acetylated GD3.

2) The CoA detected with the in vitro assay in Fig 6f could be a result of hydrolysis rather than acetyl transfer. I accept that it is difficult to quantitate the acetylated CMP-NeuNAc but it would be helpful (and feasible) to show by MS that this species is forming in an enzyme-dependent manner within the samples. This in vitro demonstration of enzyme activity would strengthen the most significant claim of the manuscript (i.e. that CASD1 is a bifunctional enzyme).

We sincerely appreciate the reviewer’s insightful suggestion. While we do agree that direct detection of O-acetylated CMP-Neu5Ac as a reaction product would be ideal, we believe that such a demonstration remains technically challenging under current experimental conditions, due to several interconnected factors:

1. Low enzyme activity resulting from limited protein yield following immunoprecipitation of full-length CASD1 from detergent-solubilized lysates, which restricts the amount of active enzyme available for in vitro assays.
2. The inherent instability of CMP-sialic acid, which undergoes rapid non-enzymatic hydrolysis to CMP and free sialic acid under the assay conditions (pH 6.5), as previously documented.
3. The extreme lability of the O-acetyl group under standard mass spectrometric conditions, which further complicates the detection of the product, particularly given the low amounts.

We believe that a definitive *in vitro* demonstration would require substantial optimization—including purification of highly active recombinant CASD1, refinement of reaction conditions (e.g., pH buffering, stabilizing agents), and the use of advanced analytical platforms. Such efforts, while scientifically valuable, represent a significant extension of the work, which we believe is beyond the scope of the current study.

Following the Editor's guidance that that demonstrating the formation of *O*-acetylated CMP Neu5Ac *in vitro* was not necessary for this revision, we have added the following statement to the Discussion Section (page23) to acknowledge it as an important direction for future work.

"Further studies are needed to directly demonstrate the *in vitro* formation of *O*-acetylated CMP-Sia by the CTMR of CASD1 and to unambiguously define the regioselectivity of the two catalytic domains of CASD1. Because acetyl group migration can mask the initial acetylation site, real-time methods such as nuclear magnetic resonance spectroscopy may be required."

We thank the reviewer for the constructive feedback and remain committed to advancing this line of research in future work.

Reviewer #3 (Remarks to the Author):

The authors have been very responsive in addressing the reviewers comments and suggestions and the revised manuscript has been greatly improved compared to the original draft. This reviewer also appreciates the difficulty of analyzing O-acetylated O-glycans in complex biological systems due to their extreme lability, which makes some experiments suggested by the reviewers unfeasible with the currently available Glycomics methodology. However, the recently released cryo-EM structure of SLC33A1 provides now additional support for the main conclusions of the manuscript. I do not have additional comments or concerns.

We thank the reviewer for their constructive comments and are pleased that the revised version satisfactorily addresses their concerns.

REVIEWERS' COMMENTS

Point-by-point response

We sincerely thank Reviewer 2 for the careful evaluation of our revised manuscript. We deeply appreciate the time and effort they devoted to reviewing our manuscript.

Reviewer #2 (Remarks to the Author)

The authors are to be commended for repeating the experiment in Fig. 5d with both UM4D4 and ICV-HE-Fc to ensure methodological consistency. This demonstrates key differences in what the detection reagents recognize. Given how important these detection reagents are in formulating the conclusions of the paper, I don't understand why the authors would want to keep it in the rebuttal only. Please add it to the SI or manuscript so the reader can appreciate the differences between these critical reagents.

We thank the reviewer for this helpful comment. The additional TLC experiment has now been included as new Supplementary Figure 14.

Regarding the *in vitro* assay, the authors state: "We believe that a definitive *in vitro* demonstration would require substantial optimization—including purification of highly active recombinant CASD1, refinement of reaction conditions (e.g., pH buffering, stabilizing agents), and the use of advanced analytical platforms." I don't disagree with this statement but isn't the point of this work to provide a definitive demonstration of the enzyme activity rather than just suggestive data?

We thank the Reviewer for this feedback. Regarding the *in vitro* assay, we followed the Editor's guidance that demonstrating the formation of *O*-acetylated CMP Neu5Ac *in vitro* was not necessary for the current revision. We aim to address this point in future work.